# Full 4D Change Analysis of Topographic Point Cloud Time Series using Kalman Filtering

Lukas Winiwarter[1,2,3], Katharina Anders[1], Daniel Czerwonka-Schröder[4,5], and Bernhard Höfle[1,6]

[1]3DGeo Research Group, Institute of Geography, Heidelberg University, Germany
[2]Integrated Remote Sensing Studio (IRSS), Faculty of Forestry, University of British Columbia, Canada
[3]Research Area Photogrammetry, TU Wien, Vienna, Austria
[4]Department of Civil and Mining Engineering, DMT GmbH & Co. KG, Essen, Germany
[5]Faculty of Geoscience, Geotechnology and Mining, University of Mining and Technology Freiberg, Germany
[6]Interdisciplinary Center for Scientific Computing (IWR), Heidelberg University, Germany

**Correspondence:** Lukas Winiwarter (papers@winiwarter.dev)

**Abstract.**

4D topographic point clouds contain information on surface change processes and their spatial and temporal characteristics, such as the duration, location, and extent of mass movements. To automatically extract and analyze changes and patterns of surface activity from this data, methods considering the spatial and temporal properties are required. The commonly used M3C2 point cloud distance reduces uncertainty through spatial averaging for bitemporal analysis. To extend this concept into the full spatiotemporal domain, we use a Kalman filter for change analysis in point cloud time series. The filter incorporates M3C2 distances together with uncertainties obtained through error propagation as Bayesian priors in a dynamic model. The Kalman filter yields a smoothed estimate of the change time series for each spatial location in the scene, again associated with an uncertainty. Through the temporal smoothing, the Kalman filter uncertainty is generally lower than the individual bitemporal uncertainties, which therefore allows the detection of more changes as significant. We apply our method to a dataset of tri-hourly terrestrial laser scanning point clouds of around 90 days (674 epochs) showcasing a debris-covered high-mountain slope affected by gravitational mass movements and snow cover dynamics in Tyrol, Austria. The method enables to almost double the number of points where change is detected as significant (from 24% to 47% of the area of interest), compared to bitemporal M3C2 with error propagation. Since the Kalman filter interpolates the time series, the estimated change values can be temporally resampled. This provides a solution for subsequent analysis methods that are unable to deal with missing data, as may be caused by, e.g., foggy or rainy weather conditions or temporary occlusion. Furthermore, noise in the time series is reduced by the spatiotemporal filter. By comparison to the raw time series and temporal median smoothing, we highlight the main advantage of our method, which is the extraction of a smoothed best estimate time series for change and associated uncertainty at each location. A drawback of the Kalman filter is that it is ill-suited to accurately model discrete events of large magnitude. It excels, however, at detecting gradual or continuous changes at small magnitudes. In conclusion, the combined consideration of temporal and spatial information in the data enables a notable reduction in the associated uncertainty of quantified change values for each point in space and time, in turn allowing the extraction of more information from the 4D point cloud dataset.

# 1 Introduction

Near-continuous time series of 3D topographic point clouds have recently become readily available through applications in research (Eitel et al., 2016), industry (Industry 4.0, e.g., Pasinetti et al., 2018), and in the public sector (e.g., disaster management, Biasion et al., 2005). Commonly, terrestrial laser scanners are installed on surveying pillars to regularly (e.g., hourly) acquire three-dimensional representations of the surrounding topography. To interpret the data for geographic monitoring, especially in terms of topographic change processes acting on the surface, information is being extracted in the form of movement patterns (Travelletti et al., 2014), objects (Anders et al., 2020), or clusters (Kuschnerus et al., 2021). This information can then be interpreted by experts to analyze change patterns and properties concerning their underlying causes, to predict future events, and to assess immediate dangers.

However, with any measurement taken in the real world, uncertainties need to be considered. In the case of topographic laser scanning, uncertainty may result in estimated change values that seemingly correspond to a change in the topography of the involved surfaces, though no real change has occurred. For example, erosion or accumulation with a low velocity is only confidently detectable after a certain period, when the change magnitude exceeds the random effects introduced by the measurements.

Two approaches can be combined to handle uncertainty: Statistical tests, such as a t-test, allow making statements about random variables by transforming them into thresholds or interval values using a confidence probability. For example, a change value of 0.01 m may have a 95% probability to be significantly different from zero. In the remaining 5% of cases, the value of 0.01 m would be caused by random errors and result in a false positive detection. The measurand (the quantity being measured) is seen not only as a single value but rather as a probability density function. An analysis of the cumulative distribution function (CDF) then gives the relation between the Type I error probability $\alpha$ (or the specificity of a test $(1 - \alpha)$) and the corresponding confidence interval. This moves the problem of change analysis or quantification to one of change detection. Such approaches have commonly been used, e.g., in tunnel deformation monitoring (Van Gosliga et al., 2006).

The other approach, alleviating uncertainty, takes advantage of the fact that no two measurements are completely uncorrelated. Generally, the closer they are to each other, the more they are alike. In space, this has been described in Tobler's first law of Geography (Tobler, 1970), and logically extends into time. In the analysis of dense time series of 3D point clouds, this fact can be used to reduce uncertainty in change detection. Consequently, lower thresholds for detecting change may be derived at the same significance probability. Change can therefore be detected as statistically significant at lower change values, or with lower change rates. To achieve this reduction of uncertainty, some sort of averaging or aggregation of multiple measurements of the same quantity into one value is required. This allows to reduce the influence of random errors, but the smoothing also reduces high-frequency signals contained in the data.

Spatial smoothing, i.e. aggregating points spatially before change analysis, reduces the spatial resolution at which change can be detected. In the widely employed multiscale model-to-model cloud comparison (M3C2) algorithm, a method to compare surfaces represented by two point clouds, a search cylinder is used to select and aggregate points of the two epochs before measuring the distance between them (Lague et al., 2013). This is beneficial over a simple cloud-to-cloud (nearest neighbor)

distance, because point clouds acquired with a laser scanner never sample the surface with the exact same pattern, and therefore no one-to-one correspondences can be established. Additionally, averaging the point positions reduces uncertainty in the surface position estimate. A more simple approach, the creation of digital elevation models of differences, also includes spatial averaging by aggregating all points within a raster cell to a single value, but is restricted to a single direction of analysis and cannot account for complex 3D topography. The variance of point distances to the fitted surfaces is typically used as a measure for the uncertainty in the estimated position in both elevation models (Kraus et al., 2006) and M3C2 distance values (Lague et al., 2013).

In the time domain, measurement series are often interpreted as signals. Signal smoothing is widely used and a multitude of methods have been established. In many approaches, a moving window is employed to aggregate multiple consecutive measurements or samples, removing or reducing outliers. Depending on the aggregation function, different filters are established, and may be mathematically described as 1D convolutions (e.g., kernel-based smoothing; Kim and Cox, 1996). Alternatively, global methods such as Fourier transform may be applied to eliminate high-frequency elements of the signal, resulting in a low-pass filter (Kaiser and Reed, 1977).

To smooth time series, (B-)splines are commonly employed (Lepot et al., 2017). Splines are piece-wise approximations of the signal by polynomial functions. Depending on the degree $n$ of the polynomials, the continuity of derivatives is guaranteed up to order $n - 1$, resulting in smooth estimates. For example, with commonly used cubic splines, the second derivative is continuous. In general, splines are interpolators, meaning they will pass through every data point. In the presence of noise, this might not be justified, and approximative splines utilizing least-squares methods have been presented (Wegman and Wright, 1983). For time series of 3D point clouds, moving average filters have been successfully used to reduce daily patterns and random effects in time series (Kromer et al., 2015; Eltner et al., 2017; Anders et al., 2019).

The geostatistical prediction method of Kriging (Matheron, 1963; Goovaerts, 1997) has been applied in the analysis of time series of geospatial data (e.g., Lindenbergh et al., 2008). Kriging allows to estimate the uncertainty of the predicted (interpolated) value to separate change signals from noise (e.g., Lloyd and Atkinson, 2001). For example, if the distance between sampling locations increases, the uncertainty for predictions between these locations will also increase, following the variogram derived in the Kriging process.

4D point cloud analyses have employed spatial and temporal smoothing separately to increase the signal-to-noise ratio of the change signal (e.g., Eltner et al., 2017; Anders et al., 2019). Kromer et al. (2015) combine both spatial and temporal neighbors of a high-frequency time series in a median filter to remove the influence of noisy observations. In this work, we present an approach that similarly combines spatial and temporal smoothing by employing a Kalman filter. In contrast to a median filter, the Kalman filter is able to consider observations having unique uncertainties, as it optimally combines these observations, and gives an estimate of the uncertainty of the result. Kalman filters are mathematical descriptions of dynamic systems and are commonly used, e.g., in navigation (Cooper and Durrant-Whyte, 1994) or traffic congestion modeling (Sun et al., 2004). Typical applications of Kalman filtering include sensor integration settings, e.g., in the integration of GNSS and IMU (inertial) measurements, when the target trajectory is smooth. A famous application was the guidance computer in the Apollo missions (Grewal and Andrews, 2010). Kalman filters are commonly used today in trajectory estimation, e.g., for

direct georeferencing of airborne laser scanning data (El-Sheimy, 2017). They have also been used for bathymetric uncertainty estimation in hydrographic applications using multi-beam echo sounding data (Bourgeois et al., 2016).

In our case, the observations are bitemporal point cloud distances, which we refer to as 'displacements' from here on out. In a Bayesian sense, each observation provides prior information on the system. The Kalman filter combines this information in a joint probability distribution to obtain estimates for the target variables that are, in general, more accurate (less uncertain) than the original observations. When estimates of position, velocity, and acceleration have been made, they can even be propagated into the future, beyond the newest measurement (Kalman, 1960).

We use the Kalman filter on change values between each epoch and a fixed reference epoch, to obtain a smoother, less uncertain time series of change for each spatial location. Accurate uncertainty estimates for the change values are obtained by applying M3C2-EP (Winiwarter et al., 2021), a method that allows the propagation of measurement and alignment uncertainties in bitemporal point cloud analysis to the obtained change values, but different methods of uncertainty quantification could be integrated. M3C2-EP contains an aggregation step derived from M3C2, where spatial neighbors are collected to create a local planar model of the surface. Using this as an input to Kalman filtering leads to spatial and temporal smoothing, where the spatial smoothing step acts as a Bayesian prior to the temporal smoothing step.

To show the applicability of our method, we analyze a synthetic scene and a dense (tri-hourly) time series of Terrestrial Laser Scanning (TLS) point clouds acquired in Vals, Tyrol (cf. Schröder et al., 2022). Near-continuous TLS surveys were carried out to ensure the safety of workers repairing the road and moving debris after a rockfall in 2017. We showcase how our method allows extracting interpretable change information from a large amount of data present in the time series of 674 epochs with about 0.6 - 1.7 million points each (after outlier removal and filtering). We further show how the smoothed time series can be used to improve the results obtained with established clustering methods, namely K-Means clustering, which has been applied to 4D point cloud data by Kuschnerus et al. (2021) to identify change patterns on a sandy beach.

The contribution of our research is twofold: First, we combine the existing method of M3C2-EP point cloud change quantification including the quantification of associated uncertainty with a Kalman filter to take advantage of the temporal domain, resulting in lower detection thresholds and less noise in the change extracted from the 3D time series. Second, we show how different smoothing methods for topographic point cloud time series influence derived change patterns in the observed scene, obtained via clustering.

## 2 Datasets

We investigate the performance of our method on two different datasets: a real scene featuring surface erosion and snow cover changes on a debris-covered slope, and a synthetic scene created from a 3D surface mesh model with known deformation properties to quantify the performance of our method.

## 2.1 Vals

For a real use case, we are using TLS data acquired over approx. three months, totaling 674 epochs from 2021-08-17 at 12:00 to 2021-11-15 at 18:00 (all times are local times) in Vals, Tyrol, Austria (WGS84: 47°02'48" N 11°32'08" E). The scene was monitored in an effort to ensure safety for the valley following a rockfall event that occurred three and a half years prior, namely on 24 December 2017. A road located immediately beneath the rockfall slope was covered in 8 m of debris, and a total rock volume of 117,000 m$^3$ was relocated (Berger et al., 2021). Data was recorded using a RIEGL VZ-2000i TLS permanently installed on a survey pillar in a shelter on the opposite slope about 500-800 m from the area affected by the rockfall.

As the point clouds are not perfectly aligned to each other (Schröder et al., 2022), retroreflective survey prisms located around the debris-covered slope were measured using RIEGL's 'prism fine scan' measurement program. These prism scans were carried out every hour in between regular, tri-hourly scans (e.g., at 13:00, at 14:00, and then again at 16:00, with point cloud acquisition of the full scene in high resolution at 15:00). The positions of the prisms were extracted following Gaisecker and Schröder (2022), using two amplitude thresholds. The angular position of the prisms was calculated from the points with the highest amplitude, whereas the ranging component was calculated based on a plane fit through the points around these maxima. Comparisons with a total station and an EDM calibration line showed accuracies for this prism detection of a few mm to < 2 cm at ranges of up to 1,200 m (Gaisecker and Schröder, 2022).

We define the epoch 2021-08-17 at 11:00 as global reference (the prism scan prior to the 'null epoch') and derive transformation parameters using the prism positions. In addition to the parameters for a 7-parameter Helmert transformation, a full covariance matrix is derived through adjustment computation. The transformations are subsequently applied to the full high-resolution scans by using the respective previous prism fine scan (i.e., prism scan one hour before full scan).

During this period, both natural, as well as anthropogenic surface changes, were captured. To investigate the benefits of full 4D point cloud analysis, we focus on relatively small-magnitude and long-duration changes. We therefore select an area of interest consisting of the debris-covered slopes, excluding the valley in which excavator works lead to sudden and high-magnitude changes (Fig. 1a). For these types of surface changes, simple bitemporal change quantification typically suffices. On 2021-11-05, heavy snowfall occurred in the area, which led to large displacement values in a short time. While most of this snow melted again by 2021-11-15, an avalanche led to a local accumulation of snow, which persisted throughout the observation period. This deposition can be seen in Figure 1b on the bottom right in red (marked as i).

The dataset presented here is part of a continuous monitoring campaign, which was operated in three subsequent setups, one in 2020 and two in 2021, and of which we use data from the third setup. It was designed to collect data for various research and development activities regarding the deployment of long-range laser scanners within a remotely controlled, web-based monitoring system from an engineering geodetic perspective. In addition to the laser scanner, inclination sensors on the pillars (PC-IN1-1° from POSITION CONTROL) and various meteorological sensors were used in the shelter. For the first two setups, a total station (LEICA TM30) and additional meteorological sensors placed throughout the area of interest were employed. The additional measurements are used to verify systematic error influences on the result results and align well with the transformation parameters extracted from the prism scans (Schröder et al., 2022).

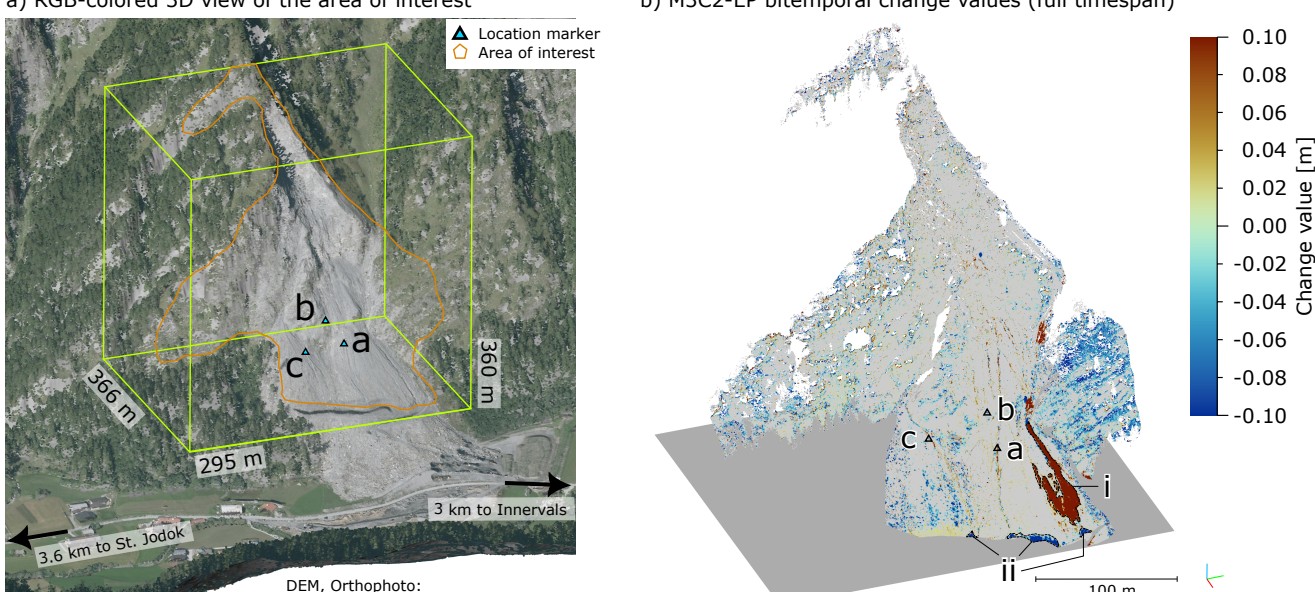

a) RGB-colored 3D view of the area of interest

b) M3C2-EP bitemporal change values (full timespan)

**Figure 1.** a) 3D perspective view of the study area (WGS84: 47°02'48" N 11°32'08" E). The debris-covered slope can be seen in gray. The point cloud which is used for analysis and which is shown in subsequent plots is outlined in orange, and single locations investigated later on are labeled as a, b, and c. b) Bitemporal displacements estimated with M3C2-EP over the full timespan. The change of > 0.1 m on the lower right is packed snow after an avalanche (i). The blue edge at the bottom is erosion due to an anthropogenic break line in the terrain (ii).

In addition to dataset alignment, preprocessing consisted of the removal of outliers and vegetation points using the statistical outlier filter (k=8, multiplier=10.0; Rusu et al., 2008) and the SMRF filter (cell size=0.5 m, slope=2; Pingel et al., 2013), as well as a filter on the waveform deviation (≤50), all implemented in PDAL (PDAL Contributors, 2018). The parameter file is supplied with the code (see *Code availability* statement).

In these data, we quantified bitemporal change and uncertainties using M3C2-EP (presented in detail in Sect. 3.1). The M3C2 distancing was carried out for a subset of the null epoch (the "core points") created by distance-based subsampling in CloudCompare, reducing the number of points to around 555,000 (min. point spacing: 0.25 m). We used the same normal vectors for all epochs, which were calculated using a 5 m search radius on the null epoch. The projection radius was 0.5 m and the maximum cylinder length was 3.0 m. With this 0.5 m search radius, we ensured that a sufficient number of points were found for the central area of interest (the debris-covered slope). Fig. 2 shows a 3D plot and a histogram of the number of points found in the cylinder for a sample epoch.

To estimate the ranging uncertainty and its variation over time, we also used the prisms installed in the scene. After extracting them from the full high-resolution scans using thresholding on the returned amplitude and approximate locations of the prisms, a planar fit was carried out. The variance of the orthogonal distances to this plane was then extracted for each prism and averaged for each epoch. The resulting precision measure, ranging from 0.004 m to 0.006 m (standard deviation), was used

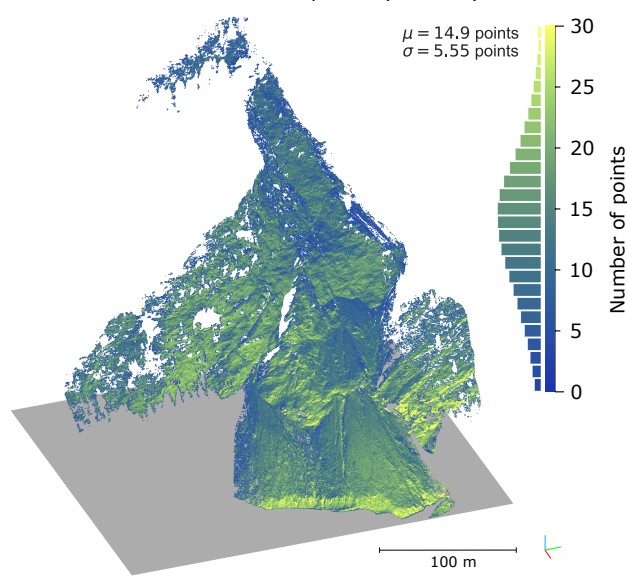

**Figure 2.** 3D view of the number of points found in the M3C2 search cylinders for epoch 2021-11-15 18:00. The search radius of 0.5 m was chosen such that most areas of interest (especially the debris-covered slope, cf. Fig. 1a) are well-represented in most epochs. The colorbar shows the distribution of the values as a histogram.

as an input for M3C2-EP for each epoch, respectively. Epochs where no prisms were detected due to precipitation were assigned the maximum ranging uncertainty value of 0.006 m. For angular uncertainty in azimuth and elevation, we used 0.0675 mrad, derived from the beam divergence as presented in Winiwarter et al. (2021) (again values of single standard deviation, respectively).

## 2.2 Synthetic scene

To validate and compare different methods of 4D point cloud processing, we created a synthetic 4D point cloud dataset. A mesh model of a 100 m×100 m plane was crafted for 40 epochs by sampling points in a regular $1 \times 1$ m grid and computing a Delaunay triangulation. Different change values were then applied over time. The magnitude of the change values ranged from 0.00 m to 0.05 m (linear gradient), and the temporal pattern was modeled by a sinusoidal function (Eq. 1). This pattern was chosen to obtain a non-uniform, yet continuous velocity and acceleration.

$$f(t) = (\sin(t) + 1)/2 \qquad \text{for } t \in [-\pi/2, \pi/2], \text{ mapped to days 0 to 40} \tag{1}$$

We applied displacements orthogonally to the mesh surface and rotated the mesh to represent a slope of $60°$. Subsequently, we performed virtual laser scanning from a single TLS position located 300 m away from the plane center (cf. Fig. 3a), using

**a) Synthetic dataset generation**

Virtual laser scanning     Augmented displacement

Virtual
VZ-400     x 40 Epochs

**b) Displacement over time**

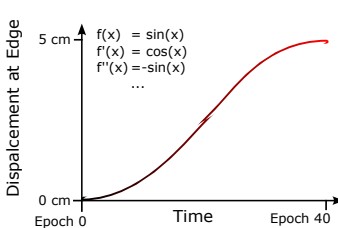

**Figure 3.** Workflow depicting the generation of the synthetic dataset: First, a planar object is created and scanned from a range of 300 m using virtual laser scanning. The planar object is subsequently deformed according to a sine function, and repeatedly scanned, resulting in a maximum displacement of $\pm$ 5 cm at the edges and 0 cm in the center of the dataset.

the specifications of a RIEGL VZ-400 TLS implemented in the HELIOS++ laser scanning simulator (Winiwarter et al., 2022). The resulting point spacing ranged from 0.63 m to 1.04 m. To simulate alignment errors, we randomly drew transformation parameters for a 7-parameter Helmert transformation from a normal distribution ($\mu = \mathbf{0}, \sigma_x = \sigma_y = \sigma_z = 0.002$m, $\sigma_\alpha = \sigma_\beta = 0.001°, \sigma_\gamma = 0.005°, \sigma_m = 0.00001$ppm) for each epoch. The uncertainty values were derived to be similar to the maximum values encountered in the real dataset.

In every epoch, a different transformation was then applied to the point cloud. Subsequently, M3C2-EP was used to quantify bitemporal surface changes and associated uncertainties, where the same normal distribution parameters were used as covariance information for the transformation. The full point cloud of the null epoch (no deformations) was used as core points, and the normal vector was defined to be the plane normal vector of the original mesh for all points. For M3C2-EP and M3C2 distance calculations, a search radius of 1 m was used, resulting in an average of 10 points falling within the search cylinder.

## 3  Methods

In this section, we

1. show how measurement uncertainties can be propagated to bitemporal change values using M3C2-EP (Sect. 3.1),

2. present a baseline method of time series smoothing using a temporal median filter (Sect. 3.2),

3. introduce the Kalman filter-smoother and the corresponding equations (Sect. 3.3), and

4. use clustering to identify areas of similar change patterns (Sect. 3.4).

The full processing workflow is shown in Figure 4.

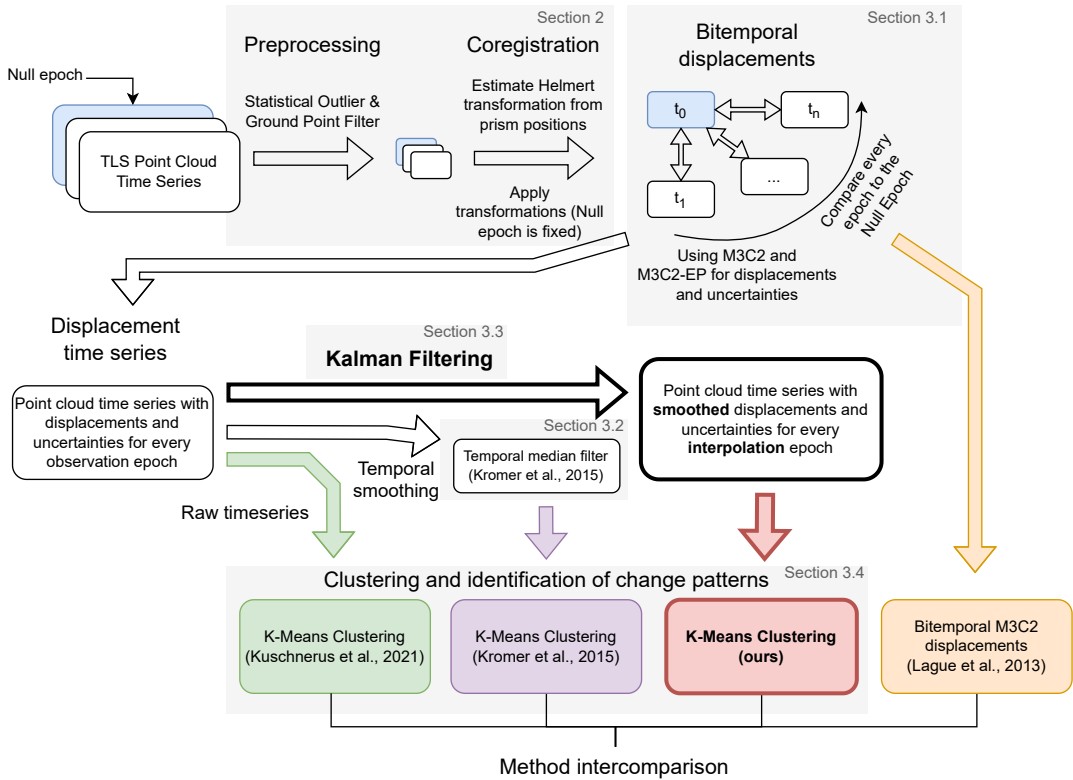

**Figure 4.** Flowchart of the workflow undertaken in this research. The novel method is highlighted using bold arrows and boxes. Three time series-based and one bitemporal method are compared. Additionally, we use K-Means clustering following Kuschnerus et al. (2021) on the multitemporal methods and compare the resulting clusters. The grey boxes refer to sections in this paper.

## 3.1 Bitemporal change analysis using M3C2-EP

To enable analysis of the time series, we convert the 4D point cloud into a series of change values at selected locations (the core points). As we want to rigorously consider uncertainties in order to separate noise from change signal, we employ multi-
scale model-to-model cloud comparison using error propagation (M3C2-EP, Winiwarter et al., 2021). This method considers measurement and alignment errors of the laser scanning observations to estimate the uncertainty of obtained change values. In turn, such uncertainties can be used in statistical tests on the significance of change values (Lague et al., 2013). Typically, the test is expressed as a spatially varying Level of Detection, which is a measure of how large a change value has to be in order to be confidently attributed to actual change. The Level of Detection depends on a significance level, which is typically set to
95% (Lague et al., 2013; James et al., 2017). In Winiwarter et al. (2021), we showed how the Level of Detection can be derived from knowledge on the sensor accuracies (e.g., from the sensor data-sheet) and alignment accuracy (using an ICP alignment; Besl and McKay, 1992) by error propagation. In addition, the individual measurements are weighted by their respective uncer-

tainties to arrive at an unbiased estimate for the change values. We refer to this method as M3C2-EP, as it extends the M3C2 algorithm by error propagation.

The error propagation is carried out by taking the mathematical model of how point cloud coordinates are obtained from transforming measured quantities (range, azimuth angle, and elevation angle) and computed quantities (transformation parameters). This model is then linearized by a Taylor approximation. Following Niemeier (2001), the uncertainty in the Cartesian target coordinates ($C_{xyz}$) can then be estimated by multiplying the linear approximation model in the form of the Jacobian matrix $A$ onto the covariance matrix of the input quantities $C_{r\varphi\theta}$ from the left, and the transpose of the Jacobian from the right,

respectively (cf. Eq. 2).

$$C_{xyz} \quad = \quad A \quad \cdot \quad C_{r\varphi\theta} \quad \cdot \quad A^T \tag{2}$$

While M3C2 itself also quantifies the uncertainty of the estimated bitemporal displacements, this estimate is inferred from the data distribution and influenced by non-orthogonal incidence angles, and object roughness within the M3C2 search cylinder (Fey and Wichmann, 2017; Winiwarter et al., 2021).

The M3C2-EP point cloud distance measure hence allows transferring uncertainty attributed to each of the original measurements, i.e., laser ranges and angular measurements, to uncertainty in point cloud displacement for every individual core point. Additionally, uncertainties from the alignment of the two datasets are considered. These uncertainties are highly correlated between points of the same epoch. Thereby, the obtained M3C2-EP distance and its spatially heterogeneous uncertainty are representing our knowledge of the point cloud displacement itself, not of the measurements. This property allows us to use the

displacement for following analyses.

We resample the time series to a regular dataset by using linear interpolation to fill in missing data points, e.g. caused by temporary occlusion in the observed scene.

## 3.2   Temporal median filter

As a baseline method to compare the Kalman filter result with, we use a temporal median filter for smoothing the time series,

as presented by Kromer et al. (2015). In this "sliding window" approach, the median function is applied to all change values in a temporal window. The median function has the advantage that the exact points in time when change is measured will not be altered. Furthermore, outliers can be completely removed, as single extrema in the input time series will not propagate to the smoothed result.

When applying error propagation on the median filter, the output value can be seen as a linear combination of the input

values multiplied by weights $w \in 0, 1$, where a single value has weight 1 and all other values have weight 0. Consequently, any uncertainty in the measurement of that single value directly propagates into the output of the median, and it is independent of the number of points in the window and the window size.

Furthermore, a window size needs to be chosen. If chosen too large, temporary surface alterations, such as a deposition of material followed by erosion, will be smoothed out. For too small windows, the benefit of smoothing in terms of outlier elim-

ination becomes negligible. To account for this, the window must be chosen smaller than the expected change rates (Kromer et al., 2015) thereby depending on the change process that is investigated. In our application, we chose 96 hours as the window size in order to remove any daily and sub-daily effects.

### 3.3 Kalman filter and smoother for change analysis

We present the use of a Kalman filter, which can be used to incorporate multiple observations (in our case the change values
for each epoch, quantified along the local 3D surface normals using M3C2-EP, cf. Section 2.1) to obtain predictions about the displacement at arbitrary points in the time series, analogous to the median smoothing. A main advantage of the Kalman filter is its potential to consider uncertainties in the inputs, allowing for observations of different qualities to be combined, and in the output. For each point in time, an uncertainty value is estimated, allowing for statistical testing of the obtained smoothed change values (as in the M3C2 for bitemporal change values). The Kalman filter is commonly employed for smooth,
continuous time series, yet not all changes are smooth in our case of 4D point cloud change analysis. The limitations arising thereof are discussed later (Section 5).

While the Kalman filter is an assimilation method, which allows updates by adding new data points, we consider an *a posteriori* analysis and assume that all measurements are available at the time of analysis. This allows us not only to consider previously observed change values at a given location, but also to incorporate future observations. To that end, we can make
use of the full 4D domain of the dataset.

The Kalman filter can be seen as a temporal extension of adjustment computation. It allows the integration of measurements over time into a *state* vector $x_t$ describing the system at a specific point in time $t$. This state can contain information on position, velocity, acceleration, or other quantities. In the following, we first present the main quantities to be defined for the Kalman filter, and then provide the prediction and update (also called correction) equations.
For the propagation from one state to the next (the "prediction" step), the state transition matrix $F$ is used[1]. It is a linear approximation of how the state changes from one point in time to the next, based on all values in the current state. The following examples of matrices and vectors correspond to the implementations of the methods presented in this paper. For a state vector $x_t$ containing the position, the velocity, and the acceleration of an object, the state transition matrix is given in Eq. 3:

$$F = \begin{pmatrix} 1 & \Delta t & \frac{\Delta t^2}{2} \\ 0 & 1 & \Delta t \\ 0 & 0 & 1 \end{pmatrix} \tag{3}$$

Here, the next position (at $t_1 = t_0 + \Delta t$) derives from the current position (at $t_0$), onto which the velocity multiplied by the time step and the contribution of the acceleration are added. The diagonal entries of Eq. 3 with value 1 ensure that the current position, velocity, and acceleration are transferred to the next point in time. Note that we present a model of order 2 here (including position, velocity, and acceleration, i.e., a quadratic term as the highest-order term in the polynomial). If velocities

---

[1]We use the nomenclature of the Python package "FilterPy" and the accompanying book "Kalman and Bayesian Filters in Python" (Labbe, 2014)

are assumed constant, allowing for an acceleration term in the Kalman filter will lead to overfitting. Therefore, models of lower order may be considered (e.g., order 1, where velocity and position are modelled, or order 0, where solely position is modelled). In the case of $F$, the respective $n \times n$ submatrix in the top left corner can be extracted for these cases (where $n-1$ is the order of the model).

An observation (a single measurement) $z_t$ may subsequently be introduced in the "correction step". For this, a linear approximation of the measurement function $H$ is required. This vector allows to project the state vector into the observation space, where the residuals of the observations are minimized. An observation consisting only of the position of the object, or in our case, the change magnitude at a position, results in a vector $H$ as shown in Eq. 4. The velocity and acceleration are not observed, so the second and third elements are zero. One could also imagine including physical measurements of velocity, e.g., using a Doppler radar system, or of acceleration, such as from an inertial measurement unit. In our application of terrestrial laser scanning repeated from a fixed position, such measurements are typically not available when investigating geomorphic surface changes. Again, when using a Kalman filter model of order $n-1$, only the first $n$ elements of the vector should be taken.

$$H = (1,0,0)^T \tag{4}$$

When considering long time series of topographic point cloud data, the local direction of the surface (here calculated as the normal vector of the null epoch) might change. A way to incorporate this into the Kalman filter would be to project the quantified changes onto the original change direction, e.g. by altering $H$ to be $H = (1/\cos(\varphi),0,0)^T$, where $\varphi$ is the angle between the inital and the updated direction. As the projection from observed value to state vector would be a multiplication by $\cos(\varphi)$, we need to set $H$ to the inverse of that projection. The angle $\varphi$ would change over time, therefore $H$ would become time-dependent as well: $H \rightarrow H_t$. For the sake of simplicity, however, we will not consider this case in the further derivations in this paper.

The step size of the update $\Delta t$ is not necessarily equal to the measurement interval, leading to prediction steps without correction steps. This allows the estimation of the state for points in time where no measurement was recorded, based on previous measurements only. Hence, state estimation is not limited to interpolation but also allows extrapolation into the future.

As in adjustment computation, every observation $z_t$ in the Kalman filter is attributed with uncertainties, herein presented by the observation noise matrix $R_t$. In our application, we use the uncertainty in point cloud distance obtained by M3C2-EP for each epoch's change value.

The process noise matrix $Q$ represents how much uncertainty is introduced in each prediction step, and therefore depends on the time step $\Delta t$. By transitioning from $t$ to $t + \Delta t$, the state vector becomes more uncertain, unless new measurements are introduced. $Q$ is representative of the system's ability or resistance to change outside of the filter constraints. We can, for example, assume a system with constant acceleration. However, in reality, this is not the case, as, e.g., in gravitational

mass movement processes, friction coefficients between topsoil and stable subsurface will change for different temperatures, moisture, and other parameters, so we allow for an adaptation of the acceleration over time.

A common approach to model process noise is discrete white noise. Here, the variance of the highest-order element (e.g., the acceleration) is defined as $\sigma_a^2$ ($\sigma_v^2$ or $\sigma_x^2$ for lower-order models). The effect of this variance on the other elements of the system's state (i.e., velocity and position) is calculated following Equation 5, (Labbe, 2014, Chapter 7). We adopt this practice for our method.

$$Q_{xva} = \begin{pmatrix} \frac{\Delta t^4}{4} & \frac{\Delta t^3}{2} & \frac{\Delta t^2}{2} \\ \frac{\Delta t^3}{2} & \Delta t^2 & \Delta t \\ \frac{\Delta t^2}{2} & \Delta t & 1 \end{pmatrix} \sigma_a^2 \tag{5}$$

The exact choice of this process noise model, especially the choice of the value of $\sigma_a^2$, is critical to the success of Kalman filtering and can be compared to the choice of the window size in the median filter. Therefore, we compare different choices for $\sigma_a^2$, investigate the resulting time series and pick one where the over-fitting of the measurement data is reduced while the model still is flexible enough to represent most of the subtle changes in the dataset appropriately. For order 1 and order 0 models, the white noise models are given in Equations 6 and 7, respectively. Depending on the choice of model, $Q_{xva}$, $Q_{xv}$ or $Q_x$ take the place of the matrix $Q$ in the following.

$$Q_{xv} = \begin{pmatrix} \Delta t^2 & \Delta t \\ \Delta t & 1 \end{pmatrix} \sigma_v^2 \tag{6}$$

$$Q_x = \sigma_x^2 \tag{7}$$

To initialize the iterative Kalman filter algorithm, starting values for the state and its uncertainty are required. As we start the time series at the null epoch with zero change, we define the initial state vector to be $x_{t_0} = (0, 0, 0)^T$. Because the change values are quantified with respect to this null epoch, we can be certain that there is no change in the null epoch (by definition) and therefore set the variance of the position to be $\sigma_{x_{t_0}}^2 = 0$ m$^2$. We allow the velocity and acceleration to take other values, and set them to $\sigma_{v_{t_0}}^2 = \sigma_{a_{t_0}}^2 = 1.0$ m$^2$/day$^2$ / m$^2$/day$^4$. In contrast to the choice of the process noise model $Q$, the impact of the exact choice of these starting values is negligible, as long as they are larger than the expected magnitude of velocity and acceleration (Labbe, 2014, Chapter 8).

Running the Kalman filter then results in estimates of the state and its uncertainty for each point in time, based on all previous states and measurements. This is referred to as a "forward pass", as calculation on a time series starts with the first measurement and then continues forward in time (Gelb et al., 1974, p. 156). The forward pass is given by the prediction (Eqs. 8-9) and update equations (Eqs. 10-13) (Labbe, 2014, Chapter 6):

First, we predict the future state $\bar{x}_{t+\Delta t}$ (at $t + \Delta t$) from the current state ($x_t$) and the linear state transition $F$. The bar on $\bar{x}_{t+\Delta t}$ signifies a prediction without an update:

$$\bar{x}_{t+\Delta t} = Fx_t \tag{8}$$

Next, we predict the future covariance by using the law of error propagation (cf. Eq. 2) and add the process noise $Q$. Here, the appropriate version of $Q$ is used, depending on the order of the model (either $Q_{xva}$, $Q_{xv}$, or $Q_x$).

$$\bar{P}_{t+\Delta t} = FP_tF^T + Q \tag{9}$$

In consequence, the state of the system becomes less certain over a longer time and can be made more certain by introducing a new observation with adequate uncertainty. For example, in the case of near-continuous TLS, change can be estimated one day into the future after having acquired one week of hourly observations. This allows estimating whether a larger interval between the observations still fully represents the expected changes.

If an observation at a given point in time $t + \Delta t$ is available, we subsequently update the state and state covariance. If not, Equations 8 and 9 are repeated for the next time step. Otherwise, for the update step, we first calculate the residuals $y_{t+\Delta t}$ by projecting the predicted state $\bar{x}_{t+\Delta t}$ into the observation space using the measurement function $H$ and subtracting this from the observations $z_{t+\Delta t}$:

$$y_{t+\Delta t} = z_{t+\Delta t} - H\bar{x}_{t+\Delta t} \tag{10}$$

Using the predicted covariance, the measurement function $H$, and the measurement noise $R_{t+\Delta t}$ (corresponding to the observation $z_{t+\Delta t}$), the so-called Kalman gain matrix $K_{t+\Delta t}$ is calculated:

$$K_{t+\Delta t} = \bar{P}_{t+\Delta t}H^T \left(H\bar{P}_{t+\Delta t}H^T + R_{t+\Delta t}\right)^{-1} \tag{11}$$

Finally, the Kalman gain $K_{t+\Delta t}$ is used to update the state vector by applying it on the residuals $y_{t+\Delta t}$ and adding to the predicted state $\bar{x}_{t+\Delta t}$:

$$x_{t+\Delta t} = \bar{x}_{t+\Delta t} + K_{t+\Delta t}y_{t+\Delta t} \tag{12}$$

Similarly, the state covariance is updated through the Kalman gain. Note that the term $KH$ is subtracted from the Identity matrix $I$, corresponding to decreasing uncertainty values:

$$P_{t+\Delta t} = (I - K_{t+\Delta t}H)\bar{P}_{t+\Delta t} \tag{13}$$

Repeating Equations 8-13 for each time increment and measurement results in a set of state vectors $x_t$ and corresponding covariances $P_t$, which are a filtered and interpolated representation of the input time series. Each state is influenced by the previous states and measurements, as well as their corresponding uncertainties.

We can, however, also include consecutive states and measurements in the estimation of the state, which can decrease uncertainty and lead to a better estimate of the state as, e.g., outliers are much more easily detected compared to just using

a forward pass. The Rauch-Tung-Striebel (RTS) smoother is a linear Gaussian method (such as the Kalman filter itself) to consider consecutive states of the system (Rauch et al., 1965). It operates backward on the time series, starting with the latest Kalman state estimate ("backward pass"). The final result is then a smoothed, estimated time series, making use of all of the available information (Gelb et al., 1974, p. 169). For discrete, instantaneous changes, the backward pass further ensures that the resulting time series will have a change of curvature temporally collocated with the change event, allowing for precise extraction of the event's timing. For more detail on the RTS smoother and its alternatives, the reader is referred to in-depth literature (e.g., Gelb et al., 1974; Labbe, 2014). In the results, we always show the RTS smoother estimates.

It is important to note that the choice of a null or reference epoch influences the results, as all change detections and quantifications relate to this epoch. The Kalman-filtered smooth trajectory and its corresponding uncertainty also signify the change related to the null epoch, and a choice of a different reference epoch would likely result in different detected changes.

## 3.4 Clustering and identification of change patterns

To represent the information contained in the time series in a static map, we use a clustering approach. Here, data points with similar features are aggregated into groups or clusters. Due to its unsupervised nature, no training data is required, which would often be lacking in the case of topographic monitoring of scenes typically featuring variable, *a priori* unknown surface dynamics. Instead, the resulting clusters can be analyzed with respect to their size, location, and magnitude, as well as visually by their shape in 3D space, and ultimately attributed to certain process types. We use K-Means clustering, which has been found to perform well for the identification of change patterns in 4D point clouds by Kuschnerus et al. (2021). As feature space, the estimated time series of change values for each core point, i.e., a list of displacement values over time, is used. The spatial component, i.e., the location of the core point in the scene, is not included in the clustering, meaning that any spatial patterns emerging in the clusters are solely due to time series similarity. We assess how the differently smoothed time series lead to different cluster results in Section 4.4.

In K-Means, the clustering algorithm iteratively minimizes the total sum of all distances from data points to the centroids of $k$ clusters (Hartigan and Wong, 1979). As the distance calculation is Euclidean, all dimensions are expected to be in the same unit and scale. An important parameter is the selection of the number of clusters. We, therefore, create segmentations with 4, 8, 10, and 12 clusters and compare them visually. The goal is to detect all groups of different processes acting on the scene while avoiding splitting up groups of the same processes into subclusters (over-segmentation).

## 4 Results

We first present the impact of different model and parameter choices on the clustering and change detection results (Sect. 4.1), and select an optimum variance value (the choice of $\sigma_x^2$, $\sigma_v^2$, or $\sigma_a^2$, hereafter summarized as $\sigma_{x/v/a}^2$) for each of the Kalman models. Subsequently, we compare our results with the raw time series, temporal median smoothing, and bitemporal M3C2-EP (Sect. 4.2) for selected locations and for the whole area of interest. To quantify the actual improvement in terms of residuals,

we present the results of the synthetic experiment in Section 4.3. Finally, clustering is carried out on the filtered time series with the best parameter settings on the real data, the results of which we show in Section 4.4.

## 4.1 Impact of model and parameter choice

We tested three different model choices for the Kalman filter and a number of parameters for each model. The different models increase in complexity and dimensions of their state vector: The first model simply tracks the displacement value itself. The assumption for this model is that the allowed variance ($\sigma_x^2$) representing how the state may change over time is sufficient to explain the displacement changes. The second model adds velocity as a component to the state vector and imposes the restriction of the variance on the velocity component (by means of $\sigma_v^2$). According to Eq. 6, the variance in the displacement value is then derived via error propagation from the variance in velocity. Finally, a model including displacement, velocity, and acceleration is created, governed by $\sigma_a^2$. This model follows a constant, gravitational acceleration, governing the surface erosion processes observed in the scene. However, this motivation assumes that friction coefficients also change continuously throughout the observation period.

For each of the three models, we experimented with the process noise variance $\sigma$. As shown in Figure 5, larger values lead to a more flexible time series model, whereas smaller values ensure a smoother temporal trajectory. For discrete events, larger values lead to ringing effects, especially with the higher-order models that include velocity and acceleration. $\sigma_{x/v/a}^2$ or the respective standard deviation $\sigma_{x/v/a}$) also influences the achievable Level of Detection: With smaller $\sigma_{x/v/a}$ values, the Level of Detection decreases, as subsequent measurements are considered to be more correlated to each other. This is reflected by the width of the error band in the plot. Generally, lower values in $\sigma_{x/v/a}$ lead to more smoothing and lower Levels of Detection.

To find appropriate values for the state variance in each of the models, the following options were investigated: For the displacement-only model (order 0), values of $\sigma_x = 0.0002$ m, 0.0005 m, 0.001 m, 0.002 m, and 0.005 m were compared (Fig. 5a). The goal was to recover processes exhibiting smooth displacements while avoiding fitting any daily patterns, which we attribute to atmospheric effects in our case. We therefore selected $\sigma_x = 0.0005$ m as the value for subsequent analyses of our example application. Note how especially in the last third of the time series, with a larger variance in the raw time series, models with larger values for $\sigma_x$ tend to fit the oscillations rather than smoothing over them. The cause for the increased uncertainty is the location of the investigated point, which is on the edge of an erosion rill. Dependent on the alignment of individual epochs, the proportion of points within the M3C2 cylinder that have changed varies, resulting in the visible oscillating pattern.

For the velocity-based model (order 1), we investigated values of $\sigma_v = 0.002$ m/day, 0.005 m/day, 0.01 m/day, 0.02 m/day and 0.05 m/day (cf. Fig. 5b). Here, we chose $\sigma_v = 0.02$ m/day to be most appropriate for our use case. In the case of the acceleration-based model (order 2), we compared values of $\sigma_a = 0.0005$ m/day$^2$, 0.001 m/day$^2$, 0.002 m/day$^2$, 0.005 m/day$^2$, 0.01 m/day$^2$, and 0.02 m/day$^2$ (cf. Fig. 5c), and selected $\sigma_a = 0.002$ m/day$^2$ to be optimal. While Figure 5 only shows the retrieved trajectories for one specific location, our investigation was carried out for four more locations, including those shown in Figure 6, and similar results were obtained for these locations (always excluding the sudden changes induced by snowfall and avalanche events).

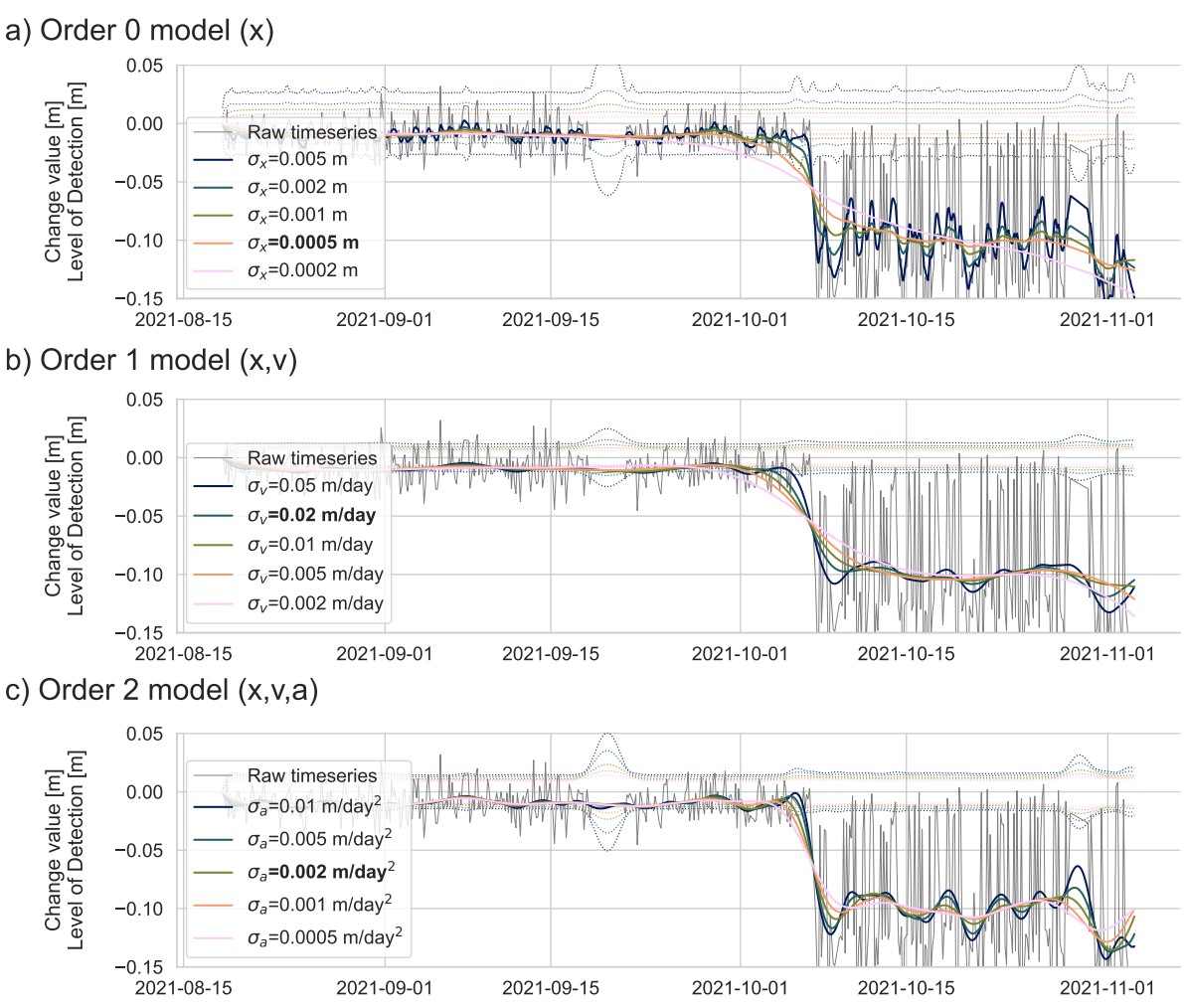

**Figure 5.** Retrieved Kalman smoother estimates and the raw time series (black) for three different models: a) using only the displacement $x$, b) using the displacement and the velocity $v$, and c) using the displacement, the velocity, and the acceleration $a$ in the model. For each model, different choices for the variance in the state vector $\sigma_x^2$, $\sigma_v^2$, or $\sigma_a^2$ were tested. The final choices (manually picked) are highlighted in bold font. The location of these time series is labeled 'a' in Fig. 1. The levels of detection at 95% significance are shown with the thin, dotted lines in the respective colors. Note how lower values of $\sigma_{x/v/a}$ lead to lower levels of detection, and more smoothing in the estimated displacement trajectories. The plots are cut off at 2021-11-03, just before the main avalanche event, as this event's magnitude exceeds the axes limits.

For sudden changes that result in a "step function"-like trajectory, smaller values of $\sigma_{x/v/a}$ strongly smooth out the trajectory of the event. While these trajectories clearly do not represent the actual change happening, the event can still be located temporally by means of the change of curvature. In Figure 5, all estimated trajectories intersect at their change of curvature points on 2021-10-05. This is a result of the minimization constraint: Assume that the trajectory before and after the change event is constant. As the RTS smoother minimizes the squared sum of residuals in observation space and constrains the change

of the trajectory (whether it be displacement, velocity, or acceleration) at the same point, the resulting best estimated trajectory must be point-symmetric around the location of maximum residuals. As continuity is required by the physical model, the change of curvature must be at that exact location. For applications, the point of changing curvature in the smoothed time series can be leveraged to temporally locate sudden changes.

    With increasing order of the model, the smooth trajectories are more oscillating. For the model of order 2 (Fig. 5c), this

can be seen especially after the sudden change on 2021-10-05. Here, a model with a comparatively high dampening ($\sigma = 0.002$ m/day$^2$) is required to avoid the oscillations, in turn limiting the ability to adapt to actual changes in the data. To further showcase the suitability of different models for different locations and corresponding change types, we present the trajectories obtained with the models of orders 0, 1, and 2, and the respective choices of state variance $\sigma_{x/v/a}$ for selected locations in the scene in Figure 6. These locations are highlighted in Figure 1. For subsequent analyses, we focus on the order 1 model, as the

order 2 model exhibits ringing artifacts at the discrete events (cf. Tekalp et al., 1989), and the order 0 model does not follow the change signal sufficiently well.

## 4.2   Comparison with other methods

To investigate the performance of the Kalman filter within the field of 4D change analysis methods, we compared our results to other methods using the same dataset (cf. Fig. 4). In Figure 6, we present (i) the raw time series of subsequent M3C2 distances,

(ii) results of temporal median smoothing following Kromer et al. (2015) with two different window sizes, and (iii) results of our method; for selected locations, which are marked in Figure 1.

    An important result of the Kalman filter is the quantification of the Level of Detection, which we compare to the Level of Detection of the bitemporal M3C2 with error propagation. In Figure 7, we show the relative number of observations (for epoch-wise bitemporal M3C2-EP with a fixed reference epoch) and the relative amount of time over the full interpolated Kalman filter

result for which the displacement magnitude is larger than the associated Level of Detection for each core point. Generally, larger values mean earlier detection of change of any type. Therein, the Kalman filter shows more changes as significant than the bitemporal approach.

    The exploitation of temporal autocorrelation, i.e., aggregating multiple measurements from multiple points in time yields much smaller Levels of Detection in the change detection, compared to the bitemporal approach. Consequently, smaller-

magnitude changes – as long as they are permanent enough for the Kalman filter to pick them up – can be detected as significant. In the case of a slope movement, this is especially important, as small movements over long timespans add up to larger displacements. Through continuous observation, the detection of surface displacement as well as the quantification of the change rate can be achieved earlier, and with higher precision.

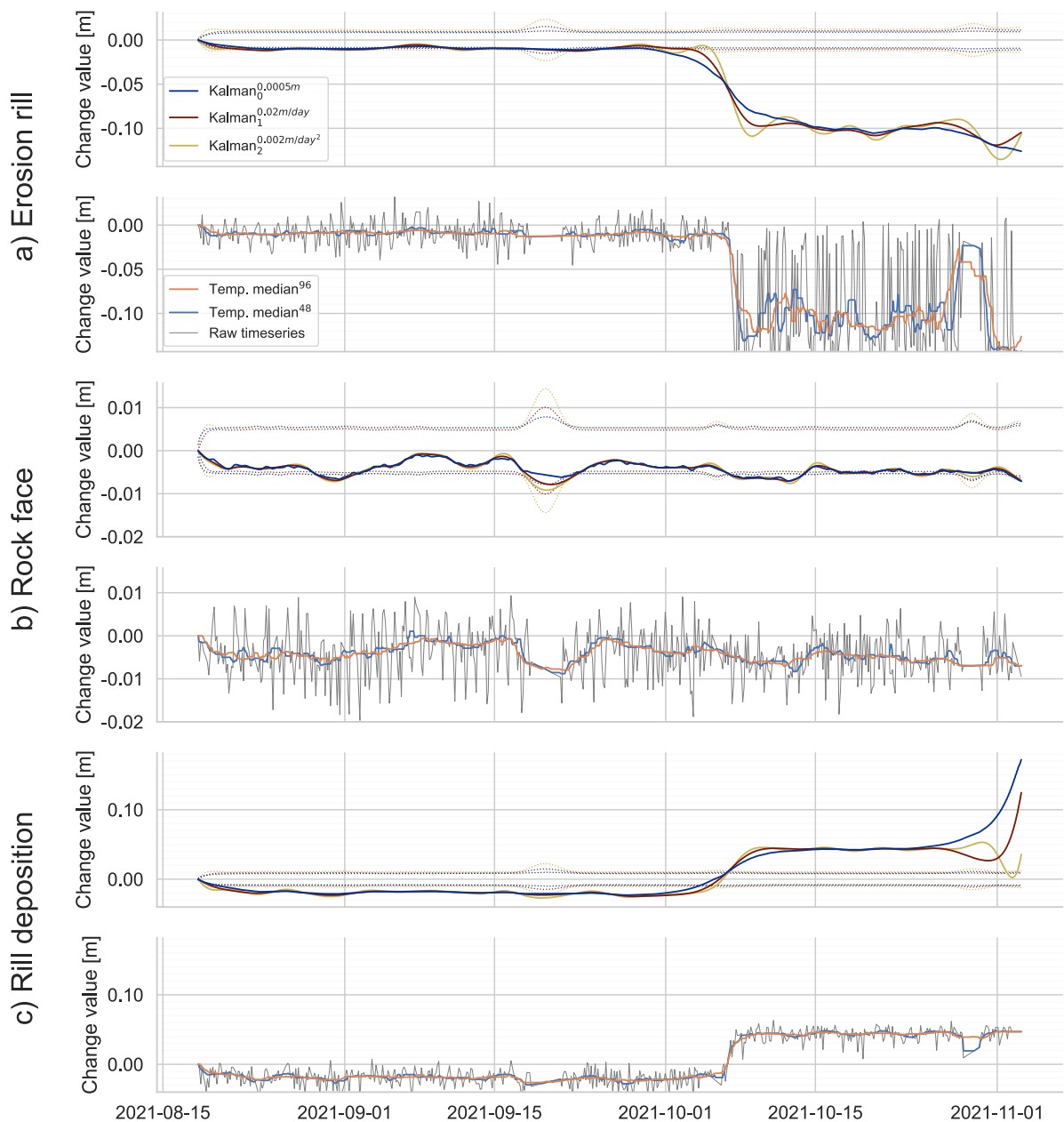

**Figure 6.** Comparison of Kalman smoother trajectories, temporal median smoothing, and the raw time series for three selected locations, corresponding to the labels in Fig. 1. In the upper plots for each location, the order 0 model is shown in blue, the order 1 model in red, and the order 2 model in yellow. The levels of detection at 95% significance are shown with the thin, dotted lines in the respective colors. The second plot for each location shows two different temporal medians (orange with a window size of 96 h, and blue with a window size of 48 h), as well as the raw time series in black. The axis limits are the same as for the first plot of each location.

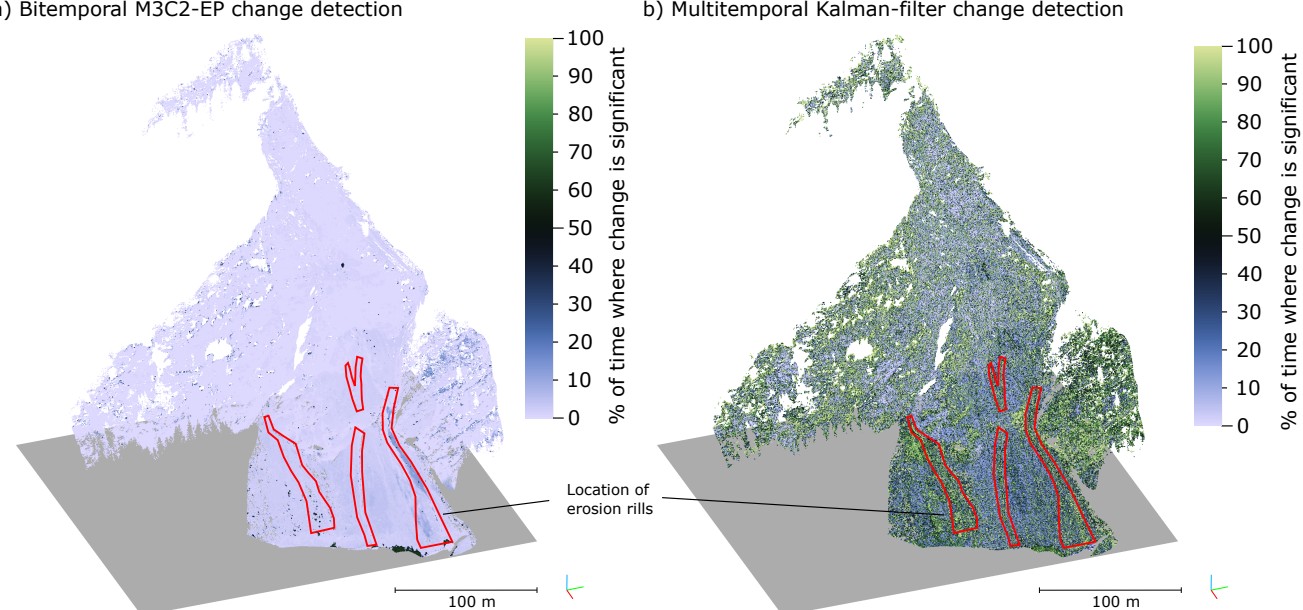

**Figure 7.** Percentage of time over the full timespan where change was detected (displacement value larger than the respective Level of Detection). The results from the bitemporal detection (a) show most points with only minimal detected change (< 10%), with a few exceptions at the bottom edge and in the avalanche-affected area on the bottom right. In comparison, the multitemporal change (b) lowers the Level of Detection so that many points, especially close to the erosion rills in the lower part of the slope (highlighted in red), show significant change over >50% of the full timespan (in shades of green).

In Figure 8a, we show the actual detected change and its magnitude for each core point at the end of the time series, i.e.,
after the avalanche event. The deposition of snow in both the avalanche area (with magnitudes > 10 cm) as well as generally in the lower slope area (with magnitudes from 2 to 6 cm) are clearly visible. In addition, significant negative change is observed on the right-hand side in blue, where vegetation is present in the dataset. Comparing Figure 8a with Figure 1b, visualizes the benefit of the multitemporal approach, as it is able to correctly detect much more change as significant than the bitemporal one.

We show the locations of points where change is detected with only the bitemporal approach, where it is detected with
multitemporal Kalman filtering, and where it is detected with both in Figure 8b. 26.92% of the core points in the study area were attributed with significant change when using the multitemporal approach but not with the bitemporal approach. This mostly concerns areas on the lower slope (colored in blue), where the magnitudes are between 0.02 and 0.06 m from deposited snow. In contrast, 4.26% of the core points were attributed with significant change in the bitemporal analysis, but not for the multitemporal case.

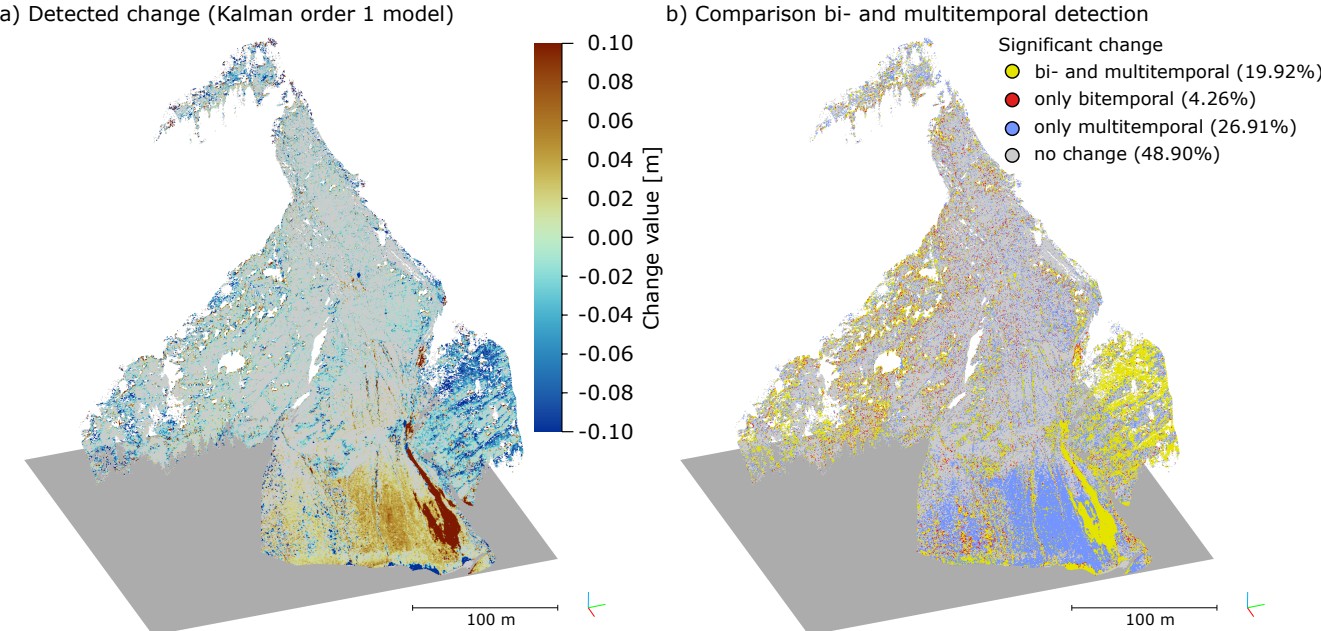

**Figure 8.** a) Change magnitudes resulting from multitemporal analysis (smoothed Kalman estimates using the order 1 model). The values displayed are the magnitudes at the end of the investigation period. Points where the magnitude is lower than the $LoDetection_{95\%}$ are colored in grey. b) Differences between bi- and multitemporal change detection. Yellow points represent locations where change has been detected as significant by both bi- and multitemporal analysis, red points are locations where change has only been detected in the bitemporal comparison, and blue points show where the multitemporal analysis enabled to detect significant change that was not detected by the bitemporal method. Subfigure b is a comparison of Subfigure a with Fig. 1b (bitemporal displacements for the last to the first epoch in the time series).

### 4.3 Results on synthetic data

For the real dataset, there is no validation data or other area-wide reference data with a much higher accuracy available, as TLS is considered to be the "gold standard". Local, point-based validation could be achieved with total station measurements, if such measurements were available within the area of interest. In our case study in Vals, total station measurements were only available for reflectors installed in stable parts outside of the area of interest. This means that we cannot investigate whether the detected change is actual change for this dataset. We therefore employed a synthetic scene with exactly known displacement to study the behavior of our method. For the analysis, we followed the same approach as with the real data, i.e., selecting a proper value of $\sigma_{x/v/a}$ for the Kalman models of order 0, 1, and 2, based on visual interpretation of the estimated trajectories. Additionally, for the synthetic data, we can quantify the residuals to the known true displacement. Table 1 shows these mean residuals for different order models and different choices of $\sigma_{x/v/a}$. In comparison, the residuals from temporal median smoothing are higher by approximately a factor of 2 and from the raw time series they are higher by a factor of 3.

| Model | $\sigma$ [m] / [m/day] / [m/day$^2$] | Sum of squared residuals [m$^2$] |
|---|---|---|
| Kalman | 0.001 | 3.636 |
| Order 0 | 0.002 | 3.154 |
| (x) | 0.005 | 4.323 |
| Kalman | 0.0002 | 3.067 |
| Order 1 | 0.0005 | 2.686 |
| (x, v) | 0.0010 | 2.923 |
| Kalman | 0.00002 | 2.864 |
| Order 2 | 0.00005 | **2.683** |
| (x, v, a) | 0.00010 | 2.797 |
| Temporal median | window size 24 | 4.297 |
| Temporal median | window size 12 | 4.435 |
| Raw time series | – | 8.425 |

**Table 1.** Sum of squared residuals (estimated - true) for the synthetic change aggregated for all core points over the full time series. The true displacement is calculated by using the y-coordinate of the core point using the model presented in Section 2.2. The minimum value is highlighted.

In addition, we show the raw time series compared to temporal median smoothing for three locations (zero displacement and large positive/negative displacement) in Figure 9. At the locations in Figures 9a and 9c, the true displacement amounts to -4.0 and +4.5 cm, respectively. Here, the Kalman filter estimates provide a much smoother trajectory than the temporal median, but tend to cut short at the curvature extremes around epochs 5 and 33, where the true change is over- or underestimated. In the case of Figure 9c, the overestimation of change is continued into epoch 40 by all methods, as measurement noise is too large. At the location shown in Figure 9b, the true change value is zero, and all derived displacement values (shown as the raw time series in black), are solely due to noise. At the end of the time series, there are a few epochs that indicate a positive change of around 3 mm, a trend which the two temporal median filters follow. The Kalman estimates, however, stay within 1 mm of the true value and are correctly below the Level of Detection.

The detected change at the end of the simulated 40-day change process is shown for all core points in Figure 10, where the different levels of detection result in a large difference in terms of detectable change. The bitemporal change detection using M3C2-EP only detects changes at the very border of the planar surface as significant, where a change magnitude > 4 cm is observed (note that the point density and change magnitude were designed specifically to demonstrate this). By using the Kalman filter on the full time series, the trajectory is smoothed and denoised, leading to a lower Level of Detection. As a result, changes above approx. 8 mm can be confidently and robustly detected. Given only the data shown in Figure 10a, it is difficult to model the true change of the underlying surface. With the data presented in Figure 10b, a spatial deformation model can be fitted, allowing to extract the real change we applied to the original mesh model with much higher precision.

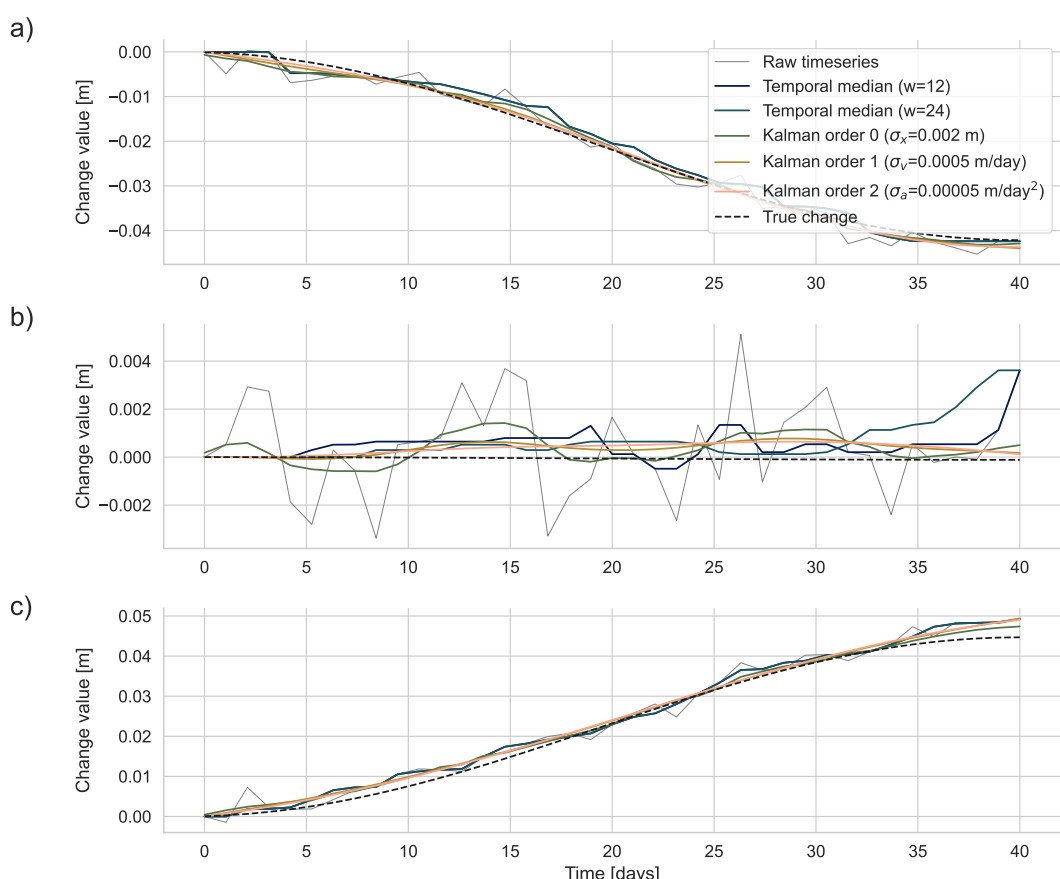

**Figure 9.** Timelines for different models representing change values in the synthetic scene. a) and c) show locations at the negative and positive extrema, respectively, and b) shows a core point at the center line of the scene, where the true displacement (black dashed line) is zero. This true displacement is calculated from the y-coordinate of the core point using the displacement formula (Equation 1).

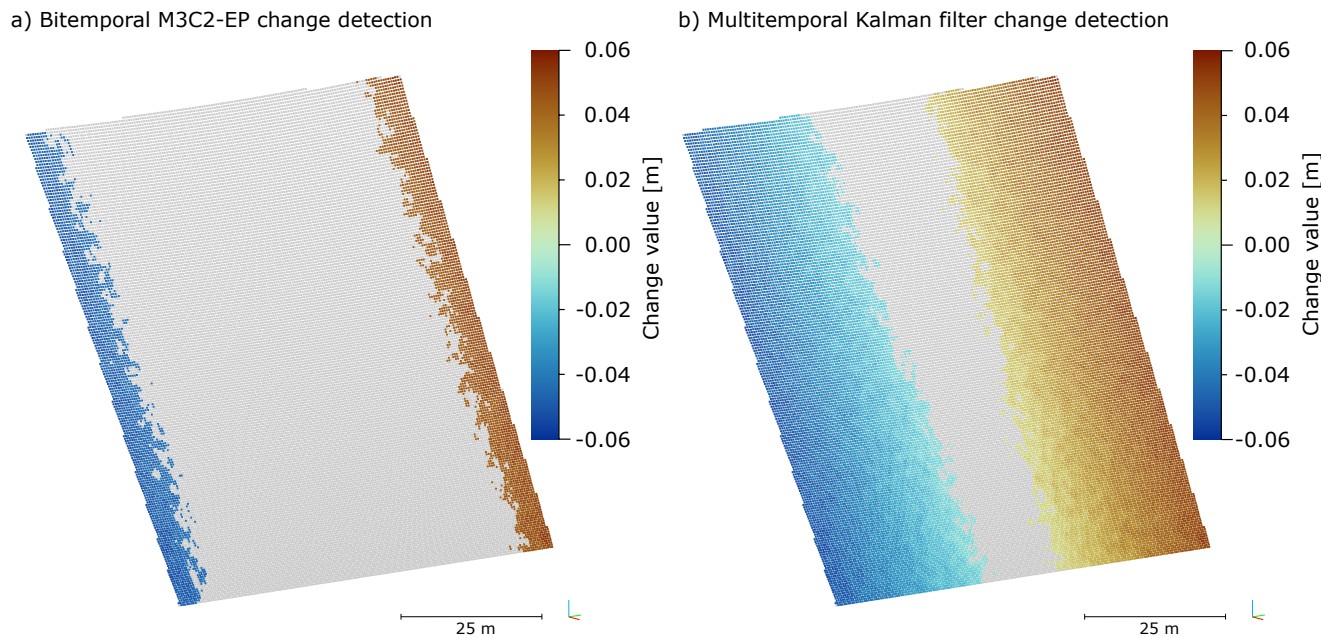

**Figure 10.** Comparison of detected synthetic change at the end of the 40-day period with a maximum displacement of $\pm 0.05$ m at the edges of the scene. The bitemporal M3C2-EP method (a) picks up changes above approx. 0.04 m, whereas the Kalman-smoother method (b) allows to detect significant changes in the time series starting from 0.008 m. Note that these changes and the error budget also include a random transformation error. From the image, it can be seen that the quantified changes are consistent, i.e., there are no changes with reversed signs on either side of the plane.

## 4.4 Clustering of change signal

To assess the influence of filtering on subsequent analyses, we use the estimated time series of change values to cluster the core points following the K-Means approach by Kuschnerus et al. (2021). As the number of clusters is an important parameter in any clustering approach, we visually inspect the results of clustering the Kalman smoothed time series (order 1 model) for 4, 8, 10, and 12 clusters (Figure 11). With increasing number of clusters, a larger number of patterns becomes visible. For our use case, we choose a cluster number of 10, and compare the results of clustering from different time series analysis methods in Figure 12.

Comparing the results from the temporal median model (using a window size of 96 hours, Figure 12a) to the Kalman filter, a less clear segmentation is observed for the temporal median model on the central slope (Cluster 8) where two erosion rills cross the segment. In the Kalman model, especially of order 1, this segment is much more clearly represented. This behavior is even more pronounced when comparing the Kalman model results to the raw time series: Here, Clusters 8 and 9 are much more mixed than in the Kalman results. Additionally, more noise appears especially in the upper half of the study site, where many green points (Clusters 6 and 7) appear. The erosion rill on the central lower slope is also less pronounced than in the other models. Still, the areas of snow accumulation are clearly segmented as Clusters 2 and 3. Note that these clusters have been created without any spatial components, i.e., spatially contiguous areas are solely the result of similar time series (i.e., change behavior).

# Result of time series clustering for different numbers of clusters

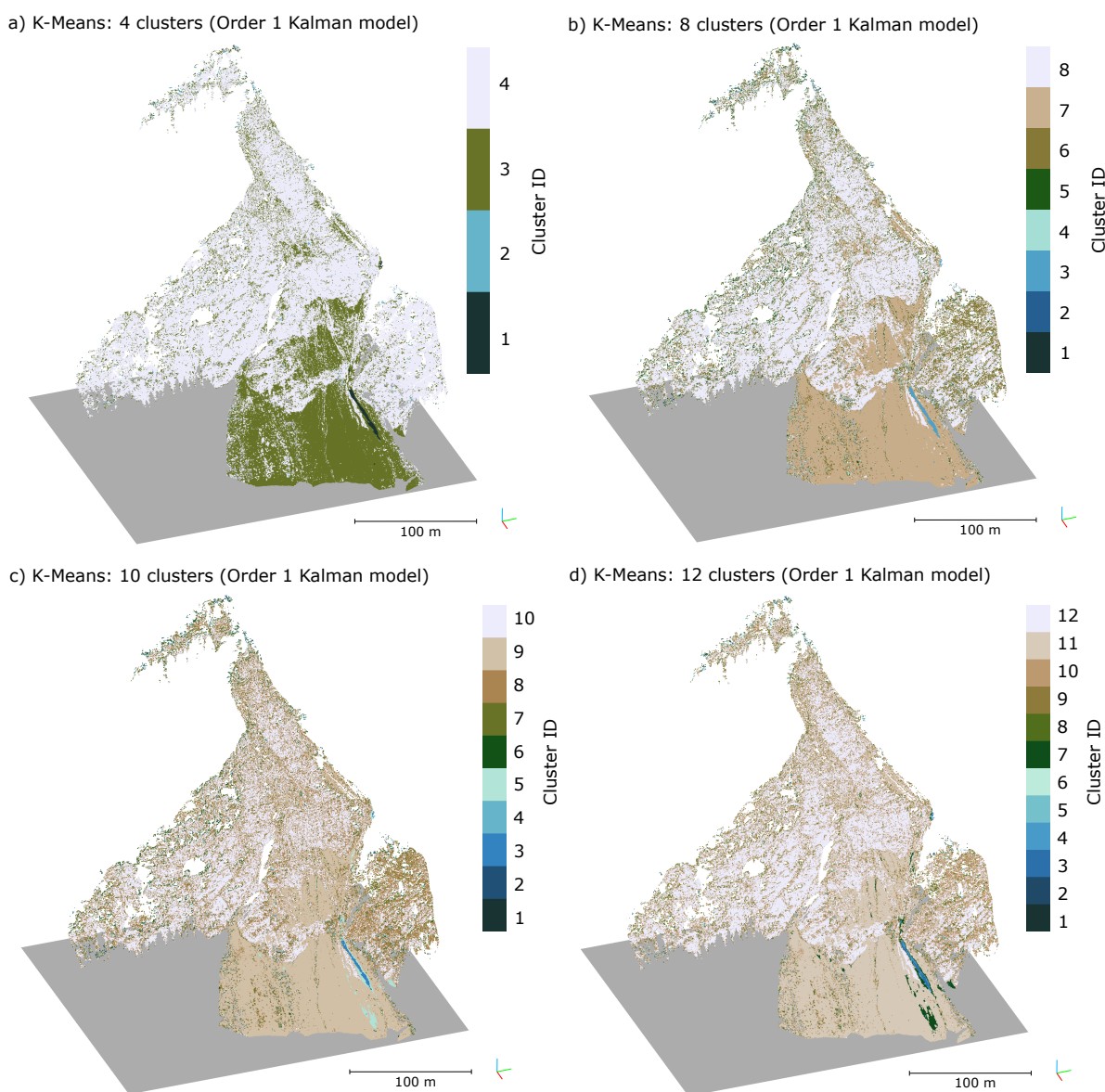

**Figure 11.** Comparison of different numbers of cluster centroids used in K-Means clustering of the Kalman smoothed time series (order 1 model, $\sigma_v$=0.02 m/day). The clusters are ordered by the number of points they contain, which results in the largest class always appearing in subtle off-white. With the addition of more clusters, patterns emerge, e.g., in the case of 10 clusters (c), the avalanche-affected area (bottom right, blue) is split into two separate segments. One of these segments (light blue) has a mean amplitude of around 0.5 m, whereas the more central one (dark blue) has a mean amplitude of 1.2 m in the Kalman filter. The segmentation further increases with 12 clusters (d).

## Result of time series clustering for different models

a) Temporal median model (K-Means: 10 clusters)          b) Raw timeseries (K-Means: 10 clusters)

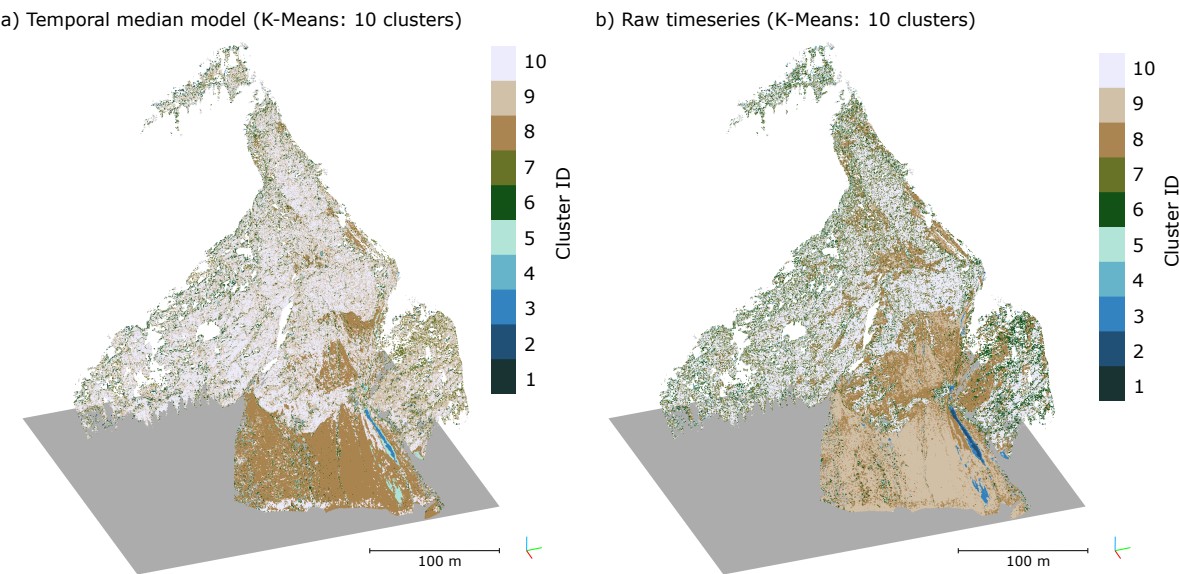

**Figure 12.** Comparison of clustering results from time series estimated using different state-of-the-art methods. a) Temporal median model (window size of 96 hours), and b) the raw time series. The cluster numbers are assigned by point count, Cluster 10 being the largest cluster.

## 5 Discussion

Kalman filtering is an alternative method for time series analysis of 3D point clouds, which, compared to the raw time series or moving median windows, rigorously considers uncertainties. As such, each observation input to the Kalman filter is attributed with an uncertainty, e.g., stemming from bitemporal change quantification using M3C2 with error propagation. This uncertainty is combined with a system state variance, a measure of how much change is expected in subsequent time periods, resulting in (a) a smoothed time series and (b) associated uncertainties. These uncertainties are not only quantified for the observation

points, but also for interpolated and extrapolated displacement values. Quantification of uncertainties allows for statistical tests of significance to separate change from noise. By analyzing the full time series instead of epoch-wise bitemporal analyses, we were able to increase the number of points where change was detected confidently at a given point in time, e.g. at the end of the time series. At our study site, the number of core points attributed with significant change was almost doubled (cf. Fig. 7). The result is confirmed by the analysis of a synthetic scene (cf. Fig. 10). The number of locations that were detected using

the bitemporal M3C2-EP method, but not when using the multitemporal approach (cf. Section 4.2), is close to the theoretical number of false positives (5% when using a level of significance of 95%), when considering that some of these 5% of false positives will again be incorrectly identified as false positives (i.e., incorrectly identified) in the multitemporal method using the same level of significance.

We compare different models by visually inspecting the estimated trajectories at sample locations (Figs. 5 and 6). In the case

of the synthetic dataset, we can further quantify a residual, as the actual change value is known, and use this to select a model order and state variance value. Here, we also showed that a properly chosen Kalman model results in a lower sum of squared residuals than the temporal median model (cf. Tab. 1). Note, however, that the synthetic change used a sinusoidal function as a model, which ensures that the changes and their derivatives in time are continuous. The Kalman filter is ill-suited to represent sudden changes, as caused by discrete events of mass movement. However, gradual motions including rockfall precursors as

studied by Abellán et al. (2009), could be detected reasonably well even without the backward pass of the RTS smoother, given that repeated observations show such a trend (Kromer et al., 2017). In such use cases, the reduction in the Level of Detection is especially crucial. Furthermore, future research could attempt to detect sudden changes in the time series, e.g., by analyzing the Kalman residuals (cf. Eq. 10) in relation to the measurement and prediction confidences.

A major challenge in the application of our method for different geographic settings is the choice of the model order (i.e.,

the physical basis) and the state variance. As no control data were available for the real dataset, we chose models by visual interpretation. We selected models that effectively reduce daily patterns, which in our data can be attributed to remaining non-linear atmospheric effects, yet do not smooth out real surface changes too much. In this study area, we select a model of order 1 for further investigations. The exact choice of model and state variance depends on the types of change processes that are being investigated. Even a spatially and temporally varying state variance could be applicable and is possible within the presented

mathematical framework. This would, however, require *a priori* knowledge of the processes acting on the surface.

In comparing the estimated Kalman trajectories to ones obtained from temporal median smoothing and to the raw time series, we demonstrate that especially with data gaps, the Kalman filter estimates often provide a more realistic interpolation

trajectory (e.g. Fig. 6c on 2021-10-28). The Kalman filter works especially well for continuous changes, and less so for discrete events. For example, in Fig. 6a and c, a discrete change occurs on 2021-10-05. The onset of this change is shifted to approx. 2021-10-03, and the target amplitude is only recovered on 2021-10-06 in the selected Kalman model of order 1. The temporal median models recover the step function much more closely here. Nevertheless, for an exact localization in time, the change of curvature of the smoothed Kalman trajectory is useful – in fact, irrespective of the choice of state variance. This can be seen in Figure 5, where all estimated trajectories intersect at this point in time.

Higher order models, especially the order 2 model, tend to overfit on step functions, resulting in ringing artifacts (blue line in Fig. 5c). The assumption for the order 2 model is that the acceleration value changes continuously, which is not fulfilled in the case of sudden, discrete change events. In the case of the order 0 model, too large choices for the state variance result in the model replicating the measurement noise (blue line in Fig. 5a). Additionally, with larger choices for the state variance, the associated uncertainties increase. A smoother model, therefore, corresponds to a lower Level of Detection. Changes that result from continuous processes acting on the surface can then be detected earlier.

As an application example, we showed how the smoothed Kalman time series can be used in K-Means clustering as presented for topographic time series by Kuschnerus et al. (2021). While there are slight differences in the results for different inputs, the main clusters are very similar for the Kalman filter methods and the temporal median smoothing. We conclude that though there are discrete changes occurring in the scene (snowfall, avalanche), which are not well represented by the Kalman filter trajectory, this does not necessarily affect the resulting clusters.

Future research could investigate how discrete change events can be identified and modeled appropriately by re-initializing the Kalman filter just after such an event. Such a re-initialization resets the estimated displacement, velocity, and acceleration (depending on the chosen order of the model), increasing the uncertainty until more observations become available and the filter converges again. In line with this consideration is the choice of uncertainty at the beginning of the process. At the start of the time series, the displacement must be – by definition – zero, and we therefore assign an uncertainty of zero to this initialization. This also ensures that all trajectories pass through the origin at the beginning of the timespan. For subsequent initializations, this argument does not hold, and a larger uncertainty (e.g., derived from the bitemporal comparison) should be assumed.

## 6 Conclusions

We presented a novel method for the analysis of 4D point clouds, supporting the monitoring of Earth surface dynamics. The application of a Kalman filter allows informed temporal smoothing, which decreases uncertainty and enables interpolation of the time series. As M3C2-EP is used to compute point cloud change values, which spatially aggregates and smooths data, the full 4D domain is exploited to find optimal estimates for change values, velocities, and accelerations. Our work can be used to detect locations and points in time where significant change occurs throughout the near-continuous 3D observation, and to group these locations into areas or subsets with similar properties. The extraction of the smoothed time series then allows the interpretation of individual trajectories where the influence of random noise is largely suppressed, which in turn allows more

precise statements about the significance of quantified change values and the properties of this change. 4D point cloud analysis using a Kalman filter and clustering techniques facilitates interpretation and allows extraction of the relevant information from the topographic point cloud time series.

The rigorous treatment of uncertainty follows a statistical approach to identify significant change and to separate it from noise resulting from sensing uncertainty and processing steps. The use of the Kalman filter thereby allows propagating uncertainties from bitemporal differencing into the time series and reduces the associated Level of Detection.

Many real-world time series datasets contain gaps or are irregular by design. With our approach, the time series can be both temporally interpolated and resampled. The regularity can subsequently be utilized by algorithms relying on a constant time step in the time series. We showed this by performing clustering of the spatial locations using the estimated change values as a

feature vector, yielding groups of similar surface change behavior.

Overall, smoothing time series while fully considering associated uncertainties is an important tool for the interpretation of topographic 4D point clouds, especially for small-magnitude changes. Such changes become especially important with increasing observation frequencies, a trend in recent near-continuous laser scanning survey setups.

*Code and data availability.*  The code used for processing the point clouds, including M3C2-EP and the Kalman filter, is available on GitHub
(https://github.com/3dgeo-heidelberg/kalman4d, v0.0.3) and is indexed with Zenodo (cf. Winiwarter, 2022). The data of the Vals debris-covered slope is available upon reasonable request to Daniel Czerwonka-Schröder at <daniel.czerwonka-schroeder@dmt-group.com>.

*Author contributions.*  **Lukas Winiwarter:** Conceptualization, Methodology, Formal analysis, Writing - Original Draft, Writing - Review & Editing, Visualization **Katharina Anders:** Methodology, Formal analysis, Data curation, Writing - Review & Editing **Daniel Czerwonka-Schröder:** Resources, Data curation, Writing - Review & Editing **Bernhard Höfle:** Conceptualization, Writing - Review & Editing, Super-
vision, Funding acquisition

*Competing interests.*  The authors declare that they have no conflict of interest.

*Acknowledgements.*  We thank the Tyrol State Government - Department of Geoinformation for their support in conducting the experimental study. We would like to thank RIEGL Laser Measurement Systems GmbH for the technical support and exchange of information during the research work. The data collection and measurement setup are supported by the European Union Research Fund for Coal and Steel [RFCS
project number 800689 (2018)]. We further wish to thank the reviewers, Dr. Roderik Lindenbergh, Mieke Kuschnerus, Dr. Dimitri Lague, and Dr. Giulia Sofia for their valuable input and critical discussion of our manuscript, as well as Dr. Fabio Crameri for his work on scientific color maps, which we have used throughout this paper (Crameri, 2021).

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
