# Peer review of "Full 4D Change Analysis of Topographic Point Cloud Time Series using Kalman Filtering"

_Earth Surface Dynamics, 2021_

## Author Comment (AC1)

**Authors' comment: ESURF-2021-103**

**Full 4D Change Analysis of Topographic Point Cloud Time Series using Kalman Filtering**

This authors' comment is a response to the reviewer comments by (1) Dr. R. Lindenbergh and M. Kuschnerus, and by (2) Dr. D. Lague. On behalf of all authors, I want to express our gratitude for taking the time to read and comment on our manuscript.

The two reviews give many helpful suggestions to improve the quality of both our research and the presentation of the results. As both reviewers indicate that the topic is of relevance to the community, we will prepare a revised version of the manuscript in the coming weeks. For now, we wish to respond to the main points of criticism (indicated as "Major comment" or "General comment" by the reviewers) and outline our revision approaches in the following.

We hope the reviewers and the editors agree that once these changes are incorporated into the manuscript, the quality is improved sufficiently for final publication in ESurf.

April 24, 2022, Lukas Winiwarter (on behalf of all co-authors)
* * *
**1. Review by Dr. R. Lindenbergh and M. Kuschnerus**

**Major comment #1:**

*The Kalman filter is one way of interpolating a time series. Directly related is Kriging, which aims at assessing and exploiting spatial and/or temporal correlation. Kriging also enables error propagation. Time series could also be approximated using Fourier or B-Spline polynomials. This could be better discussed in the Intro (as it is related work).*

We will make sure to give appropriate consideration to Kriging and Fourier/polynomial approximations in the introduction of the revised paper, as well as to discuss similarities and differences to Kalman filtering.

**Major comment #2:**

*In your Kalman filter implementation you use three parameters, location, velocity and acceleration. First it should be clearly stated somewhere that you use these parameters to model change in the vertical direction (right?). That said, using velocity and acceleration to model change at a location that changes due to digging is not directly intuitive to me, as such change would better modeled as a step function, please comment. Or more general, how should instantaneous change be incorporated in your setup? And do you really need acceleration as a third parameter, would location and velocity not provide similar results in an easier way?*

It is correct that the Kalman filter using the Newtonian model (cf. Eqs. 1 and 2) is unfit to represent sudden changes, such as those induced by the excavator works and other large episodic events including rockfalls. This is reflected in the large sum of squared residuals of the Kalman smoother results. The effect is very similar to the global test in adjustment computation, where a large residual sum can suggest an unfit functional model.

In the revision, we will add more explanation on how this sum of squared residuals could be used to detect areas/timespans where the model is unfit and how a piecewise model (re-initializing the

Kalman filter at a break point) could lead to a better estimate. Furthermore, and in accordance with comments by Dr. D. Lague, we will exclude the area of excavator work from the analysis.

Major comment #3:

*I find Figure 1 difficult to understand at first sight, would be good to also include a photo or point cloud colored by height as a fist impression of the site.*

We will add a better visualization of the study site in the revision.

Major comment #4:

*Sect. 2.5, from line 263 onwards and Sect. 2.6: Looks like you are trying many different methods at once. Why not choose one clear method of (1) pre-processing (2) uncertainties with M3C2-EP (3) smoothed time series with Kalman filter (4) feature extraction (5) clustering. Where you choose one set of (most relevant) features for the clustering. In my understanding the main goal of this paper is not to compare different features and/or clustering algorithms but to introduce the two previous steps and highlight the improvements that they yield during clustering. Possibly also add the workflow chart as shown in the readme file that comes with your code on Github. The comparison with clustering on unfiltered time series and without feature detection can then be part of the discussion.*

We agree that we might have included too many unrelated methods/results. We will revise the paper to focus on more clearly showing the benefit of (a) smoothing the time series for feature extraction (also cf. the comment by Dr. D. Lague on finite differences) and (b) kMeans clustering. We realize that the Gaussian Mixed Models are adding too many additional thoughts for a single article (weakening the focus). A workflow chart is a good suggestion and will be incorporated.

Major comment #5:

*It would be good if some or all of the features in Tables 1 to 4 could be illustrated on 2 two 3 example time series (e.g. RTS-SE-0.5) of representative locations, one in the excavation area, one in a rockfall gully and some third one.*

We will add some more representative locations instead of going into detail in the clustering. As for the features, we will survey inputs to common geophysical models (e.g., erosion models) and extract these from the time series instead of the current ones presented in Tab. 1-4.

Major comment #6:

*Results section and Figures 4-8: some subsections are needed here to make it more accessible. The first part deals with the results of processing steps 1-4. From line 300 it goes into feature extraction and comparison/visualization of different features. As mentioned above: better focus on one set of features. Then a subsection called 'comparison' is needed. Here it should be clear, what is 'your own presented method' and what other methods do you compare with. I would suggest to only compare end results and not different steps in between. Make a selection out of Figures 4 to 8 and show the most relevant results.*

This is in line with our planned revision of focussing on the kMeans clustering and removing the large number of feature-based classifications. Furthermore, this allows for a better presentation of the representative locations, the choice of the physical model, and the Kalman filter in general.

Major comment #7:

*P14, Fig 7 is hardly discussed, discuss if relevant, or omit the figure.*

We will omit this figure in the revision.
* * *
**2. Review by Dr. D. Lague**

**Major comment #1:**

*Kalman filtering is used for predicting system states that vary smoothly. I do not see why it would apply to an excavator removing rocks, rockfall or climate-driven surface erosion given that these events tends to be highly discrete in time, and thus inconsistent with a smooth evolution. Moreover, the use of a backward pass limits the ability of the method to accurately detect the timing of an event. Why can't you simply define local velocities or accelerations from a finite-difference calculation (e.g., v=(P(t2)-P(t1))/(t2-t1) where P is your point location. You'd get a better temporal localization of events, at the expense of a lower detectability of small events. As for the clustering, as you mention in the discussion, a simple linear interpolation would suffice.*

The Kalman filter is limited by the choice of functional model, which needs to be differentiable and therefore continuous. To that extent, it is true that discrete events are not well represented by the model, which is also visible in the areas of excavator work. However, while the dataset shows a rockfall-affected area, the changes that are observed represent rill erosion rather than episodic rockfalls. In the revised version, we will focus on such examples of change happing over multiple epochs, and disregard the anthropogenic change by excluding this area from the dataset.

To our understanding, the backward pass does not limit the accurate detection timing of changes. In fact, only using a forward pass would bias the detection towards later points in time (cf. the blue line in Fig. 2). This effect is removed by using the backward pass in addition. In contrast to a difference calculation, noise is effectively reduced, which is especially important if thresholding is undertaken in subsequent steps (e.g. by means of a statistical test). Taking numerical derivatives drastically increases the signal-to-noise ratio as uncertainties are amplified in the differencing. We will take care to clearly demonstrate this in the revised manuscript.

To identify the timing of events using the Kalman filter, one can, e.g., take the argument maximum of the absolute velocity (i.e., the turning points of the smoother estimate). For all choices of q, they perfectly estimate the location (cf. Fig. 3, the peak of the red curve is exactly aligned with the change event). Of course, the backward pass is only available in post-processing/offline filtering, and not fit for predictions into the future (i.e. online monitoring). We will make this distinction clearer in the revised manuscript.

**Major comment #2:**

*The part of the paper using features extracted from time series is quite superficial. There is no discussion on which features are actually important in the clustering.*

In alignment with the comments by Dr. R. Lindenbergh and M. Kuschnerus, we will remove the clustering using extracted features from the manuscript.

**Major comment #3:**

*The choice of the number of cluster is not discussed at all. This is a critical point as the issues of over or under-segmentation are critical in clustering, and not addressed at all here in the paper.*

Our experiments showed that the number of clusters used in the GMM do not have a large influence on the result in the area of the debris cone (i.e. target area). We assume this to be because of the larger variability of data (detected changes) within the surrounding forest areas, where the

additional clusters were formed. In the revised manuscript, we will take care to show these considerations and also discuss the number of clusters for the kMeans clustering.

Major comment #4:

*Some figures have poor quality, with details that are difficult to see (fig. 7,8)*

We will take care to improve the figures and include them in high quality.

Major comment #5:

*Figure 8 seems to miss 1 sub-pictures that is mentioned in the text, but not shown*

Figures will be reworked in the revised manuscript. We apologize for the incorrect reference to Fig. 8c.

Major comment #6:

*The results section does not have a clear organization, and many figure are not described and exploited to their full extent.*

As also suggested by Dr. R. Lindenbergh and M. Kuschnerus, we will thoroughly revise the results section, omit the clustering using GMM and the derived features, and focus more on the Kalman filter results as well as the kMeans clustering and differences to the state of the art. This will allow us to make better use of the more prominent results and figures.

Major comment #7:

*The discussion do not address the choice of the number of clusters. Also it does not discuss the limits of the approach, when it comes to the reduced temporal resolution, the need to choose a state variance, and the general complexity of the approach. In particular, it is ill adapted to detect precursors which can be critical in real-time monitoring because of the smoothing effect of the signal. In general, I find that there is a tendency in the discussion to assume that because the method is more complex, it is better. However, it is not clearly supported by the evidence shown in the paper. Apart from one figure (fig. 3), the superiority of the new approach compared to a simple bi-temporal approach is not clearly demonstrated (and I actually have doubts on the results of fig. 3). The benefits in terms of lower level of change detection are not obvious and would benefit from synthetic data simulation to evaluate quantitatively how each method is able to recover a known change.*

Our aim is to present the method as an extension to bitemporal change detection in the temporal domain. The notion of improving bitemporal methods is not based on the more complex model, but the increased complexity allows for the introduction of uncertainties to the analysis. We may see it as "more complex is not worse", which allows us to consider additional information in the analysis (i.e., reducing uncertainties).

The main point of doubt concerning Fig. 3 seems to be the bitemporal uncertainty quantification. We will make sure to analyse the components of uncertainty resulting from M3C2, and will present this in the updated manuscript. A first analysis showed that the LoDectection of ~8 cm at these locations is mainly a result of uncertainty in the rotation of the coordinate systems. It is yet to be determined if there is an actual rotation/tilt of the measurement pillar, e.g. due to sun irradiation, or if this is a (temperature-induced) atmospheric effect. In either case, while the changes are systematic, they cannot fully be alleviated by a rigid Helmert transformation (otherwise the ICP would solve this problem). Therefore, the residuals are treated as errors, resulting in a higher LoDetection.

In response to the comments by Dr. R. Lindenbergh and M. Kuschernus, we will extract more examples from the dataset to visualize the full time series. This will more clearly demonstrate the results of Kalman filtering at different locations. As the reviewer correctly points out, the example locations only show parts of the benefits of a Kalman filter. We therefore welcome the suggestion of synthetically altering the data. We will subsample the 4D dataset both spatially and temporally (including irregular intervals). This will result in different M3C2-EP uncertainties for different epochs, as the number of points will not be the same in each epoch (due to different subsampling). Then, we expect to show that the Kalman filter still provides an adequate fit, whereas more simple averaging or the method presented by Kromer et al. (2015) cannot use this information in the same way. With this approach, we believe that a fully synthetic dataset (e.g., by using Virtual Laser Scanning) is not required after providing clear evidence from the examples outlined above.

Finally, in contrast to simple moving window averaging of the time series, the Kalman filter combines the information about the measurement uncertainties with temporal uncertainty (as given by the q value). In consequence, the estimate of filter states between measurements is also attributed by an appropriate uncertainty value depending on how large the timespan between the surrounding measurements is. We will include these considerations in the revised manuscript.

Major comment #8:

*The discussion does not describe the benefit of Kalman filtering compared to the Kromer et al., (2015) approach.*

We will more explicitly mention the benefit of being able to use uncertainty information in connection with temporal smoothing (cf. response to previous comment).

Major comment #9:

*Also the fact that the clustering is done in 2D, while the core points are inherently 3D is not discussed.*

We guess that this is a misunderstanding. Clustering was done on the feature dimensions, and fully excluded spatial information. We will make sure to clearly explain this in the revised manuscript. Thanks for hinting to it.

Below follow two of the specific remarks we further want to address:

- *Ranging accuracy:*
  There are additional stable reflectors in the scene which were also scanned by the laser scanner. We will derive a ranging uncertainty value/model from these reflectors.

- *Comments on "discrete erosion events" and "temporal detection" vs. "prediction":*
  We understand the critique on the Kalman filter not being the best solution for discrete erosion events, and agree that the exemplary time series shown in Fig. 3 suggests we employ the Kalman filter for detection of such a discrete event. In fact, the slow but steady change after 2020-08-23 06:00 until the end of the time series much better shows the type of change the Kalman filter should be employed for.
  In the revision, we will take two steps to tackle these issues:
  1) Locate better examples in the dataset and
  2) More clearly discuss in which cases (i.e., for what types of change) Kalman filtering is an effective way to reduce the LoDetection, and when and where other methods of change detection and quantification are better suited. Moreover, we will clearly state that Kalman filtering is not "the holy grail", but rather adds to the toolset of existing 3D/4D change quantification methods.

References:

Abellán, A., Jaboyedoff, M., Oppikofer, T., and Vilaplana, J. M.: Detection of millimetric deformation using a terrestrial laser scanner: experiment and application to a rockfall event, Nat. Hazards Earth Syst. Sci., 9, 365–372, https://doi.org/10.5194/nhess-9-365-2009, 2009.

---

## Author Response (AR1)

**Authors' comment: ESURF-2021-103**

**Full 4D Change Analysis of Topographic Point Cloud Time Series using Kalman Filtering**

This authors' comment accompanies a revised version of the manuscript originally submitted on December 18, 2021. It extends the comments of the AC1 (first response to the reviewer's comments) submitted on April 25, 2022 (https://doi.org/10.5194/esurf-2021-103-AC1).

In response to the comments by Reviewer 2 (Dr. D. Lague), we have improved to coregistration procedure of the multitemporal dataset. We now use a Helmert transform derived from retroreflective prisms installed at stable positions around the area of interest. As a consequence, the derived level of detection has dropped to a point where the original dataset (acquired in 2020) started to show changes resulting from sensing errors in the data. The source of these errors, showing as missing scan lines, is unclear. However, they dominate the resulting segmentation and the change image plots, shown below:

[Figure]

Perspective view at the bottom rim of a single epoch dataset (in the direction of the arrows of the previous figure). The missing data lines are clearly visible:

[Figure]

To avoid putting a focus on these missing scan lines in the paper, we decided to use an updated dataset, acquired at the same location and in the same setting in 2021 (one year later). The dataset covers a longer period and shows multiple different surface change processes (incl. snowfall and avalanches). We believe that this adaptation is in the best interest of presenting the method of Kalman filtering for 4D change detection.

In the following, we write the reviewer's comments in *italics and grey font*, and our responses in black. We have numbered the comments for better reference. Direct quotes from the updated manuscript are highlighted in **dark green and bold font**.

We thank the reviewers for their valuable input and are looking forward that the revised, strongly improved manuscript will be considered for publication.

Aug 28, 2022; Dr. Lukas Winiwarter, on behalf of all authors

**1. Review by Dr. R. Lindenbergh and M. Kuschnerus**

1. *The presented work contains a lot of interesting ideas and visualizations and is definitely pushing information extraction from time series of 3D data a good step forward. Especially the processing steps of spatial smoothing (M3C2-EP) in combination with temporal smoothing (Kalman filter) to generate regularly sampled, smooth time series are very innovative. These smoothed time series could be used for a variety of applications and in combination with many other methods. Here the authors choose to use feature extraction and clustering to find regions of similar deformation behavior. The explanation of these last two steps lacks focus and should be concentrated on one (maximal two) sets of features and one clustering method. A separate section of the results should then deal with the comparison to other methods.*

As suggested, we have removed the calculation of engineered features and now use only kMeans clustering (following Kuschnerus et al., 2021) on the smoothed and on the raw time series data. In addition, we have included Kromer et al. (2015) as a comparison method of temporal averaging, resulting in a total of four methods to be compared: Kalman Filtering + kMeans clustering, spatiotemporal smoothing (Kromer et al., 2015) + kMeans clustering, kMeans clustering without smoothing (simple linear interpolation) and bitemporal surface change analysis. In the discussion section, we discuss the differences and similarities between the methods based on selected surface processes contained in the data.

2. *The Kalman filter is one way of interpolating a time series. Directly related is Kriging, which aims at assessing and exploiting spatial and/or temporal correlation. Kriging also enables error propagation. Time series could also be approximated using Fourier or BSpline polynomials. This could be better discussed in the Intro (as it is related work).*

We have added the following alternative time series interpolating methods to the introduction:

**To smooth observed time series, (B-)splines are commonly employed (Lepot et al., 2017). Splines are piece-wise approximations of the signal by polynomial functions. Depending on the degree n of the polynomials, the continuity of derivatives is guaranteed up to order n–1, resulting in smooth estimates. For example, with commonly used cubic splines, the second derivative is continuous. In general, splines are interpolators, meaning they will pass through every data point. In the presence of noise, this might not be justified, and approximative splines utilizing least-squares methods have been presented (Wegman and Wright, 1983). For time series of 3D point clouds, a moving average filter has been successfully used to reduce daily patterns and random effects in time series (Kromer et al., 2015; Eltner et al., 2017; Anders et al., 2019).**

**The geostatistical prediction method of Kriging (Matheron, 1963; Goovaerts, 1997) has been applied in the analysis of time series of geospatial data (e.g., Lindenbergh et al., 2008). Kriging allows to estimate the uncertainty of the predicted (interpolated) value, an important measure when attempting to separate change signals from noise (e.g., Lloyd and Atkinson, 2001). For example, when the distance between sampling locations increases, the uncertainty for predictions between these locations will also increase, following the variogram derived in the Kriging process.**

3. *In your Kalman filter implementation you use three parameters, location, velocity and acceleration. First it should be clearly stated somewhere that you use these parameters to model change in the vertical direction (right?). That said, using velocity and acceleration to model change at a location that changes due to digging is not directly intuitive to me, as such change would better modeled as a step function, please comment. Or more general, how should instantaneous change be incorporated in your setup? And do you really need acceleration as a third parameter, would location and velocity not provide similar results in an easier way?*

The Kalman filter models change in the direction of the M3C2 distances, which is not necessarily the vertical. We have clarified this in the manuscript:

**We present the use of a Kalman filter, which can be used to incorporate multiple observations (in our case the change values for each epoch, quantified along the local 3D surface normals using M3C2-EP, cf. Section 2.1) and obtain predictions about the displacement at arbitrary points in the time series, analogous to the median smoothing.**

It is correct that the Kalman filter is ill-suited for sudden changes (i.e. change occurring between successive epochs). Here, a re-initialisation of the Kalman filter could be carried out once the locations are detected. We now point this out in the discussion:

**Future research could investigate how discrete change events can be identified and modeled appropriately by re-initializing the Kalman filter just after such an event. Such a re-initialization resets the estimated displacement, velocity, and acceleration (depending on the chosen order of the model), which increases the uncertainty until more observations become available and the filter converges**

**again. In line with this consideration is the choice of uncertainty at the beginning of the process. At the start of the time series, the displacement must be - by definition - zero, and we, therefore, assign an uncertainty of zero to this initialization. This also ensures that all trajectories pass through the point at 0 at the beginning of the timespan. For subsequent initializations, this argument does not hold, and a larger uncertainty (e.g., derived from the bitemporal comparison) should be assumed.**

Furthermore, we have added a comparison with a Kalman filter model that only uses position, and one that uses velocity and position, and present the results, as well as discuss them (see new Section 4.1)

> 4. *I find Figure 1 difficult to understand at first sight, would be good to also include a photo or point cloud colored by height as a fist impression of the site.*

We have improved the visualisation of the study site in Fig. 1 following the suggestion:

[Figure]

> 5. *Sect. 2.5, from line 263 onwards and Sect. 2.6: Looks like you are trying many different methods at once. Why not choose one clear method of (1) pre-processing (2) uncertainties with M3C2-EP (3) smoothed time series with Kalman filter (4) feature extraction (5) clustering. Where you choose one set of (most relevant) features for the clustering. In my understanding the main goal of this paper is not to compare different features and/or clustering algorithms but to introduce the two previous steps and highlight the improvements that they yield during clustering. Possibly also add the workflow chart as shown in the readme file that comes with your code on Github. The comparison with clustering on unfiltered time series and without feature detection can then be part of the discussion*

In line with our response to comment #1, we have decided to completely remove the extraction of engineered features to focus on the Kalman filter as the method we are presenting in this paper. This results in a complete rework of the discussion section.

As suggested, we have further added a workflow graphic to the method section (Figure 3):

[Figure]

6. *It would be good if some or all of the features in Tables 1 to 4 could be illustrated on 2 two 3 example time series (e.g. RTS-SE-0.5) of representative locations, one in the excavation area, one in a rockfall gully and some third one.*

As we have removed the feature calculation, this comment no longer applies. However, we included more plots of smoothed time series to show how the Kalman filter deals with surface processes at different velocities (Figs. 4, 5. 6).

7. *Results section and Figures 4-8: some subsections are needed here to make it more accessible. The first part deals with the results of processing steps 1-4. From line 300 it goes into feature extraction and comparison/visualization of different features. As mentioned above: better focus on one set of features. Then a subsection called 'comparison' is needed. Here it should be clear, what is 'your own presented method' and what other methods do you compare with. I would suggest to only compare end results and not different steps in between. Make a selection out of Figures 4 to 8 and show the most relevant results.*

8. *P14, Fig 7 is hardly discussed, discuss if relevant, or omit the figure.*

We have restructured the results and discussion sections and introduced the following subsections: "Results – Impact of model and parameter choice" and "Results – Comparison with other methods". The figures have been reworked.

9. *The testing framework you mention in the 3rd paragraph of Ch.1 we applied to two epoch TLS data iDeformation Analysis of a bored tunnel by means of Terrestrial Laser Scanning, Rinske van Gosliga, Roderik Lindenbergh and Norbert Pfeifer, IASPRS Volume XXXVI, Part 5, Dresden 25-27 September 2006*

We have added the reference to the introduction section.

10. *For significant change extraction, also the terrain roughness could be incorporated as a variance value, compare Kraus, K., Karel, W., Briese, C., & Mandlburger, G. (2006). Local accuracy measures for digital terrain models. The Photogrammetric Record, 21(116), 342-354*

We have added these considerations to the introduction:

**The variance of point distances to the fitted surfaces is typically used as a measure for the uncertainty in the estimated position in elevation models (Kraus et al., 2006) and M3C2 change values (Lague et al., 2013).**

11. *In Figure 2, the velocity and acceleration could also be omitted, (or shown once, in a separate image) as the graphs have a lot of details now.*

We changed the graphs to show only change value but included different rates of smoothing into a single plot instead.

12. *Line 90: data set from 2020*

This line was removed as we changed to the 2021 dataset.

13. *Line 97: not clear what kind of comparison is meant here. Comparison of uncertainty estimation, clustering approach or change detection in general?*

We have clarified that we refer to the extracted change clusters:

**Second, we show how different smoothing methods for topographic point cloud time series influence the results of clustering to derive change patterns in the observed scene.**

14. *Figure 1: caption discusses II and III, but these are not in the figure. No reference to Fig. 1b in the text, suggested to add to section 2.3*

With the different area of interest, we changed the figures and made sure the areas are correctly labelled. Fig. 1b is now referenced:

**While most of this snow melted again by 2021-11-15, an avalanche led to accumulation of snow, which persisted throughout the observation period. This deposition can be seen in Figure 1b on the bottom right in red**.

15. *Line 124: 'methods […] are based on a part of recorded data […]' -> Methods are applied to the data, tested on the data, or similar.*

This sentence was removed during revision.

*16. Line 133 – 139: part of 2.3? not clear why it is mentioned here before M3C2-EP has been introduced.*

We believe that bitemporal M3C2-EP can be considered a preprocessing step in this analysis, and therefore opt to keep it with the data. However, we recognize that a detailed explanation of M3C2-EP is required for readers not familiar with the method (also following the request by Reviewer #2,). We therefore now explain the method in Section 2.3.

*17. Sect. 2.4: not explicitly mentioned in the text what are t and x_t*

The explanation on t and x_t have been added:

**It allows the integration of measurements over time into a state vector $x_t$ describing the system at a specific point in time $t$.**

*18. p8, r205: what exactly is the "uncertainty in point cloud distance obtained by M3C2-EP?*

We have added a more in-depth explanation on M3C2-EP, which includes the obtained uncertainties:

**This error propagation is carried out by taking the mathematical model of how point cloud coordinates are obtained from transforming measured quantities (range, azimuth angle, and elevation angle) and computed quantities (transformation parameters). This model is then linearized by a Taylor approximation. Following Niemeier (2001), the uncertainty in the target variables ($C_{xyz}$) can then be estimated by multiplying the linear approximation model in the form of the Jacobian matrix A onto the covariance matrix of the input quantities $C_{r\phi\theta}$ from the right, and the transpose of the Jacobian from the right, respectively (cf. Eq. 1).**

*19. P8r227: no variance of position in null epoch: would it not be more realistic to involve a measurement error?*

We think that an initial value of zero is appropriate for the variance in the null epoch's position, as the actual change is by definition zero, and not distributed around zero (in a Bayesian sense). We do note, however, that the Kalman filter typically takes a few epochs to converge, and that the results during this phase are less reliable. One could argue that this ought to be represented by a higher uncertainty. In reality, it does not make a difference: the filter result is bound by the first distance observation, and the uncertainty of this observation includes the uncertainty of the null epoch (as it is the uncertainty of the difference between these two epochs). We added this point to the discussion:

**[...] In line with this consideration is the choice of uncertainty at the beginning of the process. At the start of the time series, the displacement must be - by definition - zero, and we, therefore, assign an uncertainty of zero to this initialization. This also ensures that all trajectories pass through the point at 0 at the beginning of the timespan. For subsequent initializations, this argument does not hold, and a larger uncertainty (e.g., derived from the bitemporal comparison) should be assumed.**

*Figure 2: where is the example core point located in the area? You could mark the location in Figure 1, so it is visible what kind of change to expect.*

We have added markers for the locations of the displayed time series in Figure 1.

> *20. From Section 2.4 it was no clear to me why RTS was discussed, later I found out that this was actually used to obtain (additional) results, this could be better announced.*

We now better explain what the final resulting time series is in Section 3.3:

**While the Kalman filter is an "online" method, which allows updates by adding new data points, we consider a post hoc analysis and assume that all measurements are available at the time of analysis. This allows us to not only consider previously observed change values at a given location, but also future ones. That way, we can make use of the full 4D domain of the dataset.**

> *21. Is the last series of features (last paragraph of Section 2.5) necessary for this paper? In my opinion these could be omitted and focus could be on the features in Tables 1 to 4.*

We agree with the reviewer and have omitted the extraction of features, as explained with comment #1.

> *22. Figure 3b: red points (only bitemporal change) are difficult to see and it is a bit confusing that the borders of the area have the same color.*

We have improved the coloring of Figure 8b:

b) Comparison bi- and multitemporal detection

[Figure]

> *23. Caption Fig.4: -> "At grey points no significant change could be detected'*

Fig. 4 was removed from the updated manuscript.

> *24. Caption Fig. 5: 'residuals: between what and what?*

Fig. 5 was removed from the manuscript.

> *25. P16: "Fig 8 depicts a bird's eye view": this is the same view as all the other figures, and is not focusing on the lower part of the slope: wrong figure?*
> *26. P16: there is no 'II' in Fig.1.*
> *27. P16: I could not find Fig. 8c unfortunately*

We corrected the error in the manuscript layout (Figs. 7 and 8 were partially mixed up in the text due to a double identifier in LaTeX.).

> *28. P19: "Recovered velocities and accelerations": I would use the word "estimated"*

We have followed the reviewers' suggestion throughout.

> *29. P19: what do you mean by "manually extracted features"? I though all work was automated?*

Yes, the work was automated, yet the design of the features was done by hand (i.e., the selection of which features to extract). "Engineered" would have been a better term, however, we have removed the feature extraction altogether.
* * *
**2. Review by Dr. D. Lague**

> *1. while the general idea of smoothing temporally the signal to improve the signal to noise ratio and the detectability of potentially smaller events is interesting (but not new in itself, e.g., Komer et al., 2015), I find that the paper do not demonstrate clearly the benefits of the complex Kalman filtering and its associated error model compared to previous approaches (Kromer et al., 2015) or more simpler approach such as bi-temporal analysis, or simple linear interpolation when regular temporal sampling of data is needed. The paper also lacks information and discussion on key aspects of the clustering approach., and use a very complex set of features derived from time series without clear justification and in-depth analysis of the results.*

In our paper, we present the Kalman filter as an alternative approach for point cloud time series data, in line with the concept of error propagation for temporal data. In the revision, we take care to better compare the obtained results with the state-of-the-art. We therefore now compare four different methods: bitemporal M3C2, kMeans Clustering on the raw change time series, smoothed time series (Kromer et al., 2015) and ours. We are convinced that the restructured results and discussion sections show a clearer picture now.

> *2. The introduction is very good, but the result section is not well organized and many figures are not informative, or of limited quality, or not fully described in the text. A simple figure illustrating the principle of the method is also lacking.*

We have added a figure showing the principle of the method, as well as the comparisons we perform (see response to Reviewer #1, Comment #5)

Furthermore, we have strongly restructured the results and discussion.

> *3. I think it is possible for this MS to be published at some point, but it needs very significant work to better present the results (both in terms of figure quality and analysis), better demonstrate the*

*advantages of the method compared to simpler approaches, which could be done for instance on synthetic data. Also focusing the clustering approach on one method with a meaningfull set of features that would be easy to interpret would make the paper simpler.*

In line with the comments by reviewers #1, we opted to remove the feature extraction and clustering based on these features. Instead, we now focus on kMeans clustering on the time series directly, as presented by Kuschnerus et al. (2021).

We have also added a synthetic dataset to this comparison, showing the performance in the same way as for the real data.

4. *Kalman filtering is used for predicting system states that vary smoothly. I do not see why it would apply to an excavator removing rocks, rockfall or climate-driven surface erosion given that these events tends to be highly discrete in time, and thus inconsistent with a smooth evolution. Moreover, the use of a backward pass limits the ability of the method to accurately detect the timing of an event. Why can't you simply define local velocities or accelerations from a finite-difference calculation (e.g., v=(P(t2)-P(t1))/(t2-t1) where P is your point location. You'd get a better temporal localization of events, at the expense of a lower detectability of small events. As for the clustering, as you mention in the discussion, a simple linear interpolation would suffice.*

We have introduced the study site as a rockfall-affected slope, but actually focus on the surface processes acting on the debris at a much slower and less discrete pattern. To remove the dominance of the excavation works, we have removed this area from the analysis and focus on the upper slope. At the end of the 2021 time series, which we are now using, snowfall and an avalanche occur. We agree that these events are not ideally modelled by the Kalman filter, yet want to show that they *can* be represented. The discussion now includes a section on how such sudden changes, once detected, could be modelled by re-initialization of the Kalman filter state.

As already discussed in our first Authors' comment, the backwards pass does not limit temporal detection accuracy. It ensures in a global optimum (in a Bayesian filtering sense) for minimization of acceleration changes. For any discrete event, these changes will therefore be equally distributed to before and after the event, resulting in optimal detection in time. The issue with numerical differences, however, is the drastic decrease of the SNR, rendering a maxima/minima detection difficult.

We now also include clustering on the time series smoothed following Kromer et al. (2015), which allows to interpolate over data gaps. The results are compared in the discussion section.

5. *The part of the paper using features extracted from time series is quite superficial. There is no discussion on which features are actually important in the clustering.*

We have removed the feature extraction from the manuscript, following all reviewers' suggestion.

6. *The choice of the number of cluster is not discussed at all. This is a critical point as the issues of over or under-segmentation are critical in clustering, and not addressed at all here in the paper.*

In our analyses, we found that from a certain point onwards (e.g., 10 clusters), the clusters on the hill slope do not change much anymore. We assume this is due to core points in the adjacent forested areas having a higher variability, and therefore over-segmentation occurring in these areas. We have added a section on the cluster choice explaining these considerations and showing exemplary results. Note that

the cluster numbers are significantly lower in this revision than previously, as the area of interest was reduced and now excludes most forested areas.

> 7. *Some figures have poor quality, with details that are difficult to see (fig. 7,8)*

We have taken care to improve the quality of all figures.

> 8. *Figure 8 seems to miss 1 sub-pictures that is mentioned in the text, but not shown*

Unfortunately, references to Figure 8 in part actually referred to Figure 7. We have taken care to avoid this in the revision.

> 9. *The results section does not have a clear organization, and many figure are not described and exploited to their full extent.*

We have restructured the results section and improved the figures as well as their discussion.

> 10. *The discussion do not address the choice of the number of clusters. Also it does not discuss the limits of the approach, when it comes to the reduced temporal resolution, the need to choose a state variance, and the general complexity of the approach. In particular, it is ill adapted to detect precursors which can be critical in real-time monitoring because of the smoothing effect of the signal. In general, I find that there is a tendency in the discussion to assume that because the method is more complex, it is better. However, it is not clearly supported by the evidence shown in the paper. Apart from one figure (fig. 3), the superiority of the new approach compared to a simple bitemporal approach is not clearly demonstrated (and I actually have doubts on the results of fig. 3). The benefits in terms of lower level of change detection are not obvious and would benefit from synthetic data simulation to evaluate quantitatively*
> *how each method is able to recover a known change.*

As per comment #6, we have added a section on the number of clusters. In the discussion, we now include a more in-depth review of the limits of the approach and that the choice of state variance should depend on the type of observed and investigated process.

We state the limitation of pre-cursor detection in an online system, which is in line with a probability-based approach for filtering outliers.

We provide a detailed response to the results of Fig. 3 in comment #30.

Also, we have added a synthetic dataset to the comparison to show a clearer picture of the differences between the individual methods (Section 4.3).

> 11. *The discussion does not describe the benefit of Kalman filtering compared to the Kromer et al., (2015) approach.*

We have added a comparison to Kromer et al. (2015) in Section 4.2.

> 12. *Also the fact that the clustering is done in 2D, while the core points are inherently 3D is not discussed.*

As we have removed the feature-based clustering in the revision, we are now only working with kMeans clustering following Kuschnerus et al. (2021). This clustering is not spatially based, but only operates on

the time series. However, as spatially contiguous areas are subject to the same geomorphological processes, spatial clusters form.

The results of the clustering were shown in 2D views solely for visualisation purposes. We have ensured to add this critical information throughout the methods and results sections.

*13. The introduction is very good, and states clearly the objectives of the paper with thenecessary references to previous work.*

*14. L113 : please specify the typical point spacing. This is a critical information that is missing to understand why you are not able with bi-temporal analysis to detect a 5-10 cm change with a sensor with 0.005 m precision !*

We have added the point spacing information in the data section. In the original manuscript, the large uncertainty in the alignment was a result of the ICP alignment, which mainly used stable patches in the valley (i.e. closer to the scanner than the actual target area on the slope). As a result, angular uncertainties at longer ranges increased, which yielded level of detections of ~8 cm derived using M3C2-EP.

In the revised version, we are using retroreflective prisms mounted around the area of interest, avoiding extrapolation of the transformation parameters. Still, the bitemporal level of detection is at ~5 cm for many locations. This is a combination of (a) remaining alignment uncertainties, (b) ranging uncertainty (0.005 m is One-Sigma, and t-tests at p<0.05 are approximately Two-Sigma) and (c) angular uncertainty resulting from the beam divergence. The footprint (One-Sigma) at 800 m range is approximately 5.4 cm in radius. At an incidence angle of 30 deg (typical for the study area), 2.7 cm remain just from the angular uncertainty.

*15. L127 : could you explain why you needed to realign the data if the sensor was on a fixed pillar, and arguably, all scans were acquired in the same reference frame ? or is it specifically related to using M3C2-EP and estimating the alignment uncertainty ? In that case, mention it in the text.*

Similar to TLS time series from fixed positions in, e.g., Kromer et al. (2017), Williams et al. (2018), and Anders et al. (2019), we also observe that our data is not perfectly aligned, even though the sensor was installed on a fixed pillar. Especially with the lowered level of detection through temporal smoothing, the effects from imperfect alignment become relevant. With the now-included prism extraction, we can show the course of the transformation parameters over time, which clearly shows (a) daily (temperature-dependent) patterns and (b) a general increase of the yaw rotation (i.e., the axis of the main screw fixing the TLS to the pillar):

[Figure]

(for the order of rotations, refer to Joeckel, R., Gruber, F. J. (2020). Formelsammlung für das Vermessungswesen. Germany: Springer Fachmedien Wiesbaden.)

The maximum effect of singular rotations ($\alpha$ at the end of the time series approx. 0.005 gon) at 800m range corresponds to 6.3 cm tangential shift. For more information on the prism extraction, refer to Gaisecker, Schröder (2022): RIEGL V-Line Scanners for Permanent Monitoring Applications and integration capabilities into customers risk management (http://www.riegl.com/uploads/tx_pxpriegldownloads/Whitepaper_RIEGL_DMT.pdf, last accessed 2022-07-22).

Above are the actual transformation parameters – the corresponding uncertainties are shown here, where the maximum rotational uncertainty of 2.5e-5 gon corresponds to 3/10 mm at 800 m range.

[Figure]

We choose a search radius that allows for meaningful statistics (n ~ 15) in most parts of the dataset (see also the updated Figure 2 showing the point count in the M3C2 cylinder).

[Figure]

a) Distribution of number of points

b) Point count in M3C2-EP search cylinder (r=0.5 m)

*17. L138 : are there any correction for temperature effects on ranging measurement (that start to be significant over 800 m) ? Also, I suspect the 0.005 m ranging accuracy is certainly not at 800 m distance ! have you better constrained on the actual ranging accuracy at 800 m ?*

Linear dependent temperature effects are considered by including a scale parameter in the Helmert transformation (cf. Williams et al., 2018), which we use. The TLS manufacturer only provides 0.005 m as a ranging accuracy value ("tested under RIEGL conditions"). Additionally, there have been some studies investigating the accuracy of TLS over longer ranges (e.g., Fey and Wichmann, 2016), who have noted that the increase in ranging uncertainty mainly stems from the finite footprint and non-zero incidence angles.

To get an estimate of the actual ranging uncertainty, we employed a data-driven approach: For each retroreflective prism, we selected points within certain amplitude thresholds. We then calculated the quality of a plane fit to these points: As the prism represents the highest-energy reflector in the footprint for many points, the ranging component should be the same – and any deviations from it, i.e., from the best fitting plane, are due to ranging uncertainty. We applied this method for every epoch separately, resulting in a histogram of ranging uncertainties (over the different prisms) for each epoch:

[Figure]

We now use the mean values of ranging uncertainty derived for each epoch as the ranging error component in the error propagation. The obtained values are all very close to, or even below, the nominal

accuracy of 0.005 m. Any linear effects in scale, which are not well obtainable through this approach, will be covered by the Helmert transformation.

Overall, it is important to consider here that at 800 m range, the influence of the angular uncertainty is about an order of magnitude larger than the one of the ranging uncertainty.

> 18. *Section 2.1 : could you give an estimate of the mean point density of the scans ?*

Point density strongly varies with range and incidence angle – we therefore have added a figure showing the point count in the M3C2 search cylinder (Fig. 2b), which also shows this variance.

> 19. *Section 2.2 seems like an introduction to the algorithm you present to analyses PC series, with a bit of state of the art in spatio-temporal clustering. Then subsequent section (M3C2-EP etc…à) should be sub-section on this one (2.2.1, 2.2.2….) otherwise section 2.2 by itself is not part of the method.*

We have restructured the data and methods sections accordingly.

> 20. *L174 : while I know M3C2-EP, I suspect it would help less specialist readers to have a bit more explanation on the extra steps needed for the uncertainty calculation in M3C2-EP, and the benefits compared to the standard uncertainty model of M3C2. No need to go into too much detail, but the M3C2-EP paper being a tough one to read, it would help to have a self consistent paper*

We have gladly added more introduction to M3C2-EP in the method section:

**This error propagation is carried out by taking the mathematical model of how point cloud coordinates are obtained from transforming measured quantities (range, azimuth angle, and elevation angle) and computed quantities (transformation parameters). This model is then linearized by a Taylor approximation. Following Niemeier (2001), the uncertainty in the target variables ($C_{xyz}$) can then be estimated by multiplying the linear approximation model in the form of the Jacobian matrix A onto the covariance matrix of the input quantities $C_{r\phi\theta}$ from the right, and the transpose of the Jacobian from the right, respectively (cf. Eq. 1).**

**[…]**

**While M3C2 itself also quantifies the uncertainty of the estimated bitemporal differences, this estimate is derived from the data and influenced by non-orthogonal look angles, and object roughness within the M3C2 search cylinder (Winiwarter et al., 2021).**

> 21. *L179 : you should specify how k is going to be defined, as it needs to be manually chosen for k-means clustering.*

We have added a section (Section 4.4) on the choice of cluster classes (cf. response to comment #6)

> 22. *L180 : I'm roughly familiar with Kalman filtering owing to airborne LiDAR data processing, however, I suspect many readers won't, and they may have trouble following this part. Maybe a sketch of the basis of kalman filtering applied in your specific case would help.*

We have added a paragraph on the basic idea of Kalman filtering to the methods section:

**We present the use of a Kalman filter, which can be used to incorporate multiple observations (in our case the change values for each epoch, quantified along the local 3D surface normals using M3C2-EP, cf. Section 2.1) and obtain predictions about the displacement at arbitrary points in the time series, analogous to the median smoothing. A main advantage of the Kalman filter is its potential to consider uncertainties both in the inputs, allowing for observations of different qualities to be combined, as well as in the output. Here, an uncertainty value for each point in time can be estimated, which allows for statistical testing of the obtained smoothed change values (as in the M3C2 for bitemporal change values).**

**Typical applications of Kalman filtering include sensor integration settings, e.g. in the integration of GNSS and IMU (inertial) measurements, when the target trajectory is smooth. A famous application was the guidance computer in the Apollo missions (Grewal and Andrews, 2010). Kalman filters are commonly used today in trajectory estimation, e.g. for direct georeferencing of airborne laser scanning data (El-Sheimy, 2017). In our case of 4D point cloud change analysis, not all changes are smooth. The limitations arising thereof are discussed later (Section 5).**

*L220 : see major comment 1. I really have trouble reconciling the smooth nature of Kalman filtering with the highly discrete nature of erosion events*

We agree that the Kalman filter is not ideal for discrete events, cf. our response to comment #1. We discuss this limitation in the revised manuscript (Section 5) but also point out that there is no apparent effect in further processing (i.e., clustering).

23. *L241 : this sentence is not clear to me. How do you turn the 4D data into 2D ?-> ok I get it, it's an introduction to the subsequent section. Maybe rephrase to make things clearer.*

We have rewritten the section on feature extraction and do not focus on the creation of 2D maps further.

24. *L254-258 : making sense of the attributes in relation to the expected geomorphic processes would be great. For instance, it is not obvious at this stage why the total curvature is importante (compared to a more straightforward measure such as cumulative change) ?*
25. *L263 : FFT on a signal which is have periodic pattern does not really make sense especially if you're not detrending the signal and using filters to account for the finite dimension of the time series. Maybe theres's a reason I don't see, but in that case it seems important to give a little intuition as to why you suggest such features. Wavelet analysis might make more sense as it combines temporal location (when an event happen) and frequency analysis (~ duration of an event), but it's hard to come up with simple integrative features to be used for subsequent clustering.*
26. *L275 : I do not see at all, how the clustering based on the features, which are potentially very numerous and do not contain any relation to "physics" or "drivers" of cliff erosion (precipitation, local cliff geometry, ….) can actually lead to a more "physical interpretation" than analyzing the estimated change directly. The authors need to back this statement.*

We have removed extraction of engineered features as we think it would go beyond the scope of the now extended paper – thereby comments #25-27 are solved.

*27. L277 : you should mention that the number of clusters need to be specified, in case nonspecialist readers think that unsupervised clustering is just pushing a button and getting a result. An you should explain here, how you choose the number of clusters (as you did for GMM. It's critical.*

Solved with our response to comment #6.

*28. L293-299 / figure 2 : the description of the figure needs to be improed. You're first sentence stating "appropriately filters daily effects" gives a sense that 0.005 m/day² is initially the best value, while indeed you choose 0.05 m/day². Also for such an important parameter, your search of the optimum is rather qualitative. I don't think plotting acceleration helps at all. You do not discuss the occurrence of clear oscillations in the signal prior to the change. Are they real signal, or variations of the scanner position (+-2 cm, that's huge) ? It seems that another criteria for choosing σ is that it must be large enough to not trigger a detection for these oscillations.*

With the new dataset and the improved alignment, we have chosen new examples to present in our manuscript. They demonstrate that a value of 0.005 m/day^2 does appropriately filter these daily effects. Unfortunately, it also removes parts of the signal, and has large residuals around the 2020-08-22 12:00 mark. The second choice, 0.05 m/day^2 still removes the daily signal, whereas the third choice (0.5 m/day^2) overfits the daily noise.

In the updated manuscript, using different locations allows us to better discuss these effects in much more detail.

*29. Fig3 : It is hard to tell without having the information on the point cloud spacing, but I'm extremely surprised that a bi-temporal analysis is not able to detect change in the channels where distances more than 5 cm are measured by the multitemporal approach. It might be that the ICP registration has an issue on the two epochs used for testing for significant change and translated into a large registration error increasing the LoD. But 5 cm over a few cm² should be detected easily with a sensor with 1 cm ranging error (an estimate at 800 m) and a 1 cm registration error. It's very odd.*

See our response to comments #14 and #17. The positional error of sensed points are between 2 and 3 cm (std. dev., due to the footprint) and the alignment error contributes another 0.5 cm (see #15). The alignment uncertainty needs to be considered as systematic, whereas the positional error can be considered as random. With 30 points in the search radius, the resulting level of detection (according to the Equation given in Lague et al., 2013) is 2-3 cm, which is in line with the current results. Applying bitemporal M3C2 gives (on average) levels of detection between 1 cm and 10 cm (mean = 6 cm) when assuming 0.5 cm alignment error.

*30. Fig 4b: use also greyed color for the area with non significant change to facilitate comparison with 4a. It would be interesting, following fig. 3 to show if the onset of change detection differs significantly from the bi-temporal approach compared to the multitemporal approach. This would better emphasize the interest of your method.*

We have added a plot that shows the relative amount of time of the whole time series where significant change is detected for both the Kalman approach and bitemporal M3C2 (Figure 7):

[Figure]

The Kalman approach detects different surface-change inducing processes as significant, therefore a plot showing the difference in time between the first detection of significant change does not show the desired result. In the plot below, the bitemporal M3C2-EP mainly detects the snowfall event 79 days after the start of the time series (orange), whereas the Kalman approach detects small-magnitude change processes much earlier (i.e., within the first few days).

[Figure]

Left: point in time (in days) of first significant change using bitemporal M3C2-EP, right: point in time (in days) of first significant change using Kalman filtering (order 1 model, σ=0.02m/day)

We opted to include only the first of the discussed plots in the manuscript, as we believe the second one being misleading due to different change processes being detected.

*31. L318 : which "value" ? it's not clear*

The "value" referred to the RTS smoother residuals. We have removed the figure in response to comment #33.

*32. L321 : I fail to grab the interest in showing fig. 5. What do you learn and how important it is ?*

Without more in-depth analysis of the different surface processes acting on the terrain, and with the removal of the area of anthropogenic change from the dataset, Fig. 5 becomes less informative and obsolete. We have removed this part from the analysis.

*33. L325 : Ok, figure 6 tells us you use 50 clusters in one case and 100 in the second case, but it is not even mentioned in the text and you do not justify your choices. It's a critical point to discuss. Also why can't you simply create a linear interpolation between two epochs to fil in the gap for your clustering ? it would solve your problem of temporal spacing without having to rely on a complex Kalman filtering.*

We have added a section on the choice of cluster numbers (see our responses to comments #6 and #21)

*34. Question: is there correspondence between the large pink area and the area where no change is detected ?*

Referring to Fig. 6, the large pink area is closely resembling the area where no significant change has occurred. These results have changed in the revision.

*35. L330 & fig 7 : this description of figure 7 is insufficient. It is not up to the reader to analyze the results. You must highlight much more key results, otherwise it means the figure is useless (actually I'm not convinced it's actually useful, because the quality of the visualization is poor, and we don't know why you 150 clusters and not a lower or larger number).*

The clustering approach based on engineered features was removed from the manuscript.

*36. L334 : the whole subsequent section is really hard to follow.*
*37. L340 : How dependent are the so-called "distinct" features on the number of cluster. As you are using a large number of cluster, you are artificially producing many features. But this may simply result from over-segmentation. Here the choice of your number of cluster should be discussed in depth.*

Obsolete, as we have removed this section and the clustering analysis using engineered features.

*38. Figure 8 : the visualization is extremely poor, and this figure is not really usable.*

This figure was removed and we have reworked the figures in the updated manuscript to improve readability.

*39. L343 : Fig 8c is missing*

Unfortunately, some references to Fig. 7 were incorrectly labelled Fig. 8. We have thoroughly checked the updated version for such errors.

> 40. *L345 : this last statement seems to contradict previous sentences in the very same paragraph. So in the end, your method detect the same things than the others. What is really its interest beyond interpolating slightly (which could simply be done with linear interpolation between 2 epochs…) ?*

We have more clearly pointed out the benefits of filtering for time series analysis, especially with respect to error propagation. Kalman filtering is – to date – the only known time series analysis method incorporating geodetic error propagation through time. These are results from the comparisons to the other methods on both the synthetic and the real dataset.

> 41. *L355 : smoothing of the time series is debatable advantage, as it decreases the temporal resolution of event detection. Also "predicting" future states when it comes to natural environments seems hardly feasible, especially when considering rockfalls or rain-related erosion which are by nature not really predictable.*

We have carefully revised the discussion to shift the focus to (a) post-event analysis and (b) prediction of the development of uncertainty for non-discrete events. We agree that the Kalman-filter is ill-formed to detect rockfall precursors:

**The Kalman filter is ill-suited to represent sudden changes, as caused by discrete events of mass movement. However, gradual motions such as rockfall precursors as studied by Abellán et al. (2009), could be detected well even without the backward pass of the RTS smoother, given that repeated observations show such a trend. In such use cases, the reduction in the Level of Detection is especially crucial.**

> 42. *L362 : velocity and acceleration are meaningfull for estimating a plane trajectory as it by nature smooth, however it is not useful, and probably not desirable, for interpolating the occurrence of discrete erosion events.*

We agree with the reviewer that this is the case for discrete events, but it is in fact useful when investigating gradual movements, i.e. changes represented in the time series over multiple epochs. We discuss how such types of surface activities could be represented in the Kalman filter in Section 5.

---

## Referee Report (RR1)

**Review**

esurf-2021-103
Title: Full 4D Change Analysis of Topographic Point Cloud Time Series using Kalman Filtering
Author(s): Lukas Winiwarter et al.
MS type: Research article
Iteration: Revised submission
Reviewers: Roderik Lindenbergh and Mieke Kuschnerus

**Summary**

This article presents methodology to extract changes from extensive (hundreds of epochs) time series of 3D topographic data obtained by a permanently installed terrestrial laser scanner. It does so by running a Kalman filter on individual time series of spatially smoothed bitemporal point cloud distances at many different spatial 'core' locations. The Kalman filters are used to smooth the time series, to reduce uncertainty by averaging, but also to incorporate uncertainty in the change analysis. This procedure allows detecting changes that are smaller than just based on the raw time series. Smoothed time series are further clustered using k-means clustering. The new methodology is demonstrated both on a simulated data set and on a real data set of an active landslide.

**Major Points:**
- M3C2 distances are quantified perpendicular to the surface. What is the consequence of a changing surface orientation in (potentially) long time series?. Could the orientation be somehow incorporated in the Kalman filter?
- What is the consequence of using a reference epoch? Are the changes detected in, say, the 2nd half of time series independent of the choice of a reference epoch?
- F2b) suffices, no need to also show F2a;
- On the other hand, would be good to add one visualization of the synthetic data set to S2.2
- Ch. 3: the wording in the points 1-4, the section names 3.1 to 3.4 and the wording in F3 are all different, for clarity please use more similar wording and show more clearly in F3 where the steps 1 to 4 occur.
- Eq. (2) is generally known as the "error propagation law" and holds for any linear relation between observed and estimated parameters, compare, "Tiberius, C. C. J. M., van der Marel, H., Reudink, R. H. C., & van Leijen, F. J. (2021). Surveying and Mapping.", p69, https://textbooks.open.tudelft.nl/textbooks/%20catalog/book/46
- Therefore, error propagation can also be applied to the median filter, S3.2, when the smoothed value is obtained as a linear combination of existing values (with uncertainties) within the sliding window.
- The explanations on page 12 could be made more clear:
  - R278 introduces a measurement noise matrix 'R', but this matrix is not used in the consecutive explanation. Would be good to see 'R at work' in some equations.
  - R280 introduces Q, in the following, $Q_{xva}$, $Q_{xv}$ and $Q_x$ are used, but never properly introduced.
  - Similarly, $\sigma_a^2$, and $\sigma_v^2$ are never introduced (in general: explain all symbols)
  - R287: mentions $\sigma^2$ and refers to Eq. (6), but there is no $\sigma^2$ (without subscript) in Eq. (6).
  - What is the relation between 'discrete white noise" (R 286) and the following?
  - What do you mean by 'highest order"? Order in the sense of polynomials?

- In the presentation of the results, I would not present "linear interpolation" as (yet) another method, but as the raw time series
- F4: would be good to also include the original time series ("linear interpolation"). In addition, I cannot see any dotted lines in my A4 color print-out, the level of detection is not visible in the print out.
- Extend the explanation of 'reduced level of detection' due to Kalman filter approach (seems to be an important result)
- P368: intersection at "change of curvature" point: bug or feature? Is this good or bad to have?
- P376: please add a reference for the ringing artifacts.
- I feel for F5 and F6, three different locations should be enough: a) is similar to e) and somehow to d); that would also allow to merge F5 and F6, by showing the results now in F5 and F6 for say, a) c) d) together (directly below each other).
- 398: "gold standard": for local validation the Total Station data could be used. Future work could also include the Kalman predicted displacement for next epoch with the measured displacement (as another way of validation)
- F9 and F10 are shown but hardly discussed, do they really contribute to the manuscript?
- Looks like there is no link from the text to Fig. 8. Either remove the figure or discuss it (I would suggest to discuss it)
- In general, further focus your results, e.g. by elimination: First, choose your best settings for the time series processing; Second, show the improvement in change detection between best settings and traditional pair-wise M3C3, Finally, show some different results for k, but only using the best settings; The results of the simulated data set could also be moved to the discussion -> are both Figure 11 and 12 needed? Could be combined into one, showing the most interesting results
- Would be interesting to see the average Kalman velocity (acceleration) per core point.
- I would move the percentages of change locations (4.26 % in r 441) to the results
- "The Kalman filter is ill-suited to represent sudden changes": possible future work could be to test if a new measured value fits the Kalman trend given the confidence in both the trend and the measured value, this could result in a kind of trend change detection,
- Kalman filtering has been previously used to smooth bathymetric MBES data, e.g. Bourgeois, B. S., Elmore, P. A., Avera, W. E., & Zambo, S. J. (2016). Achieving comparable uncertainty estimates with Kalman filters or linear smoothers for bathymetry data. *Geochemistry, Geophysics, Geosystems*, *17*(7), 2576-2590.

**Minor Points:**
- Abstract: "almost double", relative to what baseline method?
- Abstract: "this can be a critical': can be positive or negative, maybe reformulate as 'This is a solution for subsequent analysis methods that…"
- Intro, r28: "needs to" -> "is" (needs to would require a mathematical proof that there is no other way)
- Intro, r48: "detectable" -> "detecting"
- Caption, F2a): -> "Histogram of the number of points found"
- S3.3., r250: I would call it an "assimilation" method, rather than an "online" method.
- S3.3, r250: "post hoc" -> "a posteriori"
- R219: 'right' -> left
- R247: 'empliyed'
- R259: -> "For a state vector $x\_t$ containing…"

- R263: -> The diagonal entries of Eq. (3) ensure"
- R280: skip "Finally"
- R305: -> The impact of the exact choice"
- R328: -> "time series similarity"
- R328: -> "How the differently smoothed time series" (right?)
- R331: -> "Euclidean"
- Ch 4: in the first lines, I would explicitly introduce S4.1 to s 4.4
- caption , F10: "larger" -> "starting from"

---

## Referee Report (RR2)

Comments on esurf-2021-103,
Full 4D Change Analysis of Topographic Point Cloud Time Series using Kalman Filtering

**Minor remarks:**
1. r42: singular → single (singular values belong to Linear Algebra)
2. r242: → 'number of points' (remove 'the')
3. r266: → 'implementations of the methods'
4. r324: noide → noise
5. r452: 'so long' → 'as long'
6. Caption, Figure 6: 'temproral' → 'temporal'
7. Caption, Fig. 6: → 'window size' (remove 's')
8. r462: 'either' → 'both'
9. Caption, Figure 9: 'negative maxima': is that not the minimum?
10. Caption, Fig. 10: 'still reliable' → 'consistent'

---

## Author Response (AR2)

**AUTHOR'S COMMMENTS**

Dear Dr. Lindenbergh, dear Ms. Kuschnerus,

thank you for taking the time to review our revised manuscript titled "Full 4D Change Analysis of Topographic Point Cloud Time Series using Kalman Filtering". We have taken your comments into account and believe that this iteration has substantially increased the quality of the paper, especially regarding the presentation of the results and the discussion. Please find our point-by-point response below.

On behalf of all authors,

Lukas Winiwarter

**DETAILED RESPONSE TO REVIEWER COMMENTS**

MAJOR POINTS:

M3C2 distances are quantified perpendicular to the surface. What is the consequence of a changing surface orientation in (potentially) long time series?. Could the orientation be somehow incorporated in the Kalman filter?

> That is a very interesting idea. A changing orientation of the direction of analysis using the Kalman filter is currently not possible. However, change values derived from different orientations may be projected onto a fixed direction of analysis in the update step, by adapting the H vector. Say the angle between the fixed direction and the vector of a specific epoch's M3C2 value is $\varphi$, the H vector would change to

$$H = \begin{pmatrix} 1/\cos\varphi \\ 0 \\ 0 \end{pmatrix}$$

> We have included this idea in the explanation of the H vector in Section 3.3:

> When considering long time series of topographic point cloud data, the local direction of the surface (here calculated as the normal vector of the null epoch) might change. A way to incorporate this into the Kalman filter would be to project the quantified changes onto the original change direction, e.g. by altering $H$ to be $H = (1/\cos(\varphi), 0, 0)^T$, where $\varphi$ is the angle between the inital and the updated direction. As the projection from observed value to state vector would be a multiplication by $\cos(\varphi)$, we need to set H to the inverse of that projection. The angle $\varphi$ would change over time, therefore H would become time-dependent as well: $H \rightarrow H_t$. For the sake of simplicity, however, we will not consider this case in the further derivations in this paper.

What is the consequence of using a reference epoch? Are the changes detected in, say, the 2nd half of time series independent of the choice of a reference epoch?

> The reference epoch serves as a 'zero-point' for all analyses, including the statistical test with the null hypothesis that the change between the **reference epoch** and an **epoch at time t** is zero. Therefore, the change quantification and detection directly depend on the choice of the reference epoch. We have highlighted this at the end of Section 3.3:

> It is important to note that the choice of a null or reference epoch influences the results, as all change detections and quantifications relate to this epoch. The Kalman-filtered smooth trajectory and its

corresponding uncertainty also signify the change related to the null epoch, and a choice of a different reference epoch would likely result in different detected changes.

F2b) suffices, no need to also show F2a; On the other hand, would be good to add one visualization of the synthetic data set to S2.2

We have removed Fig. 2a and included the histogram with the color scale of Fig. 2b (now Fig. 2):

[Figure]

Furthermore, we have added a schematic visualisation of the synthetic dataset as Fig. 3:

Ch. 3: the wording in the points 1-4, the section names 3.1 to 3.4 and the wording in F3 are all different, for clarity please use more similar wording and show more clearly in F3 where the steps 1 to 4 occur.

The wording is now uniform over the points in Chapter 3, the section titles, and Fig. 3 (now Fig. 4). We have also included the indices of the points 1-4 in Fig. 3 (now Fig. 4). The titles now read:

3.1 Bitemporal change analysis using M3C2-EP
3.2 Temporal median filter
3.3 Kalman filter and smoother for change analysis
3.4 Clustering and identification of change patterns

We also updated the wording of differences/distances/displacements to be uniform, and opted for "displacements".

The updated Figure is shown below:

Eq. (2) is generally known as the "error propagation law" and holds for any linear relation between observed and estimated parameters, compare, "Tiberius, C. C. J. M., van der Marel, H., Reudink, R. H. C., & van Leijen, F. J. (2021). Surveying and Mapping.", p69, https://textbooks.open.tudelft.nl/textbooks/%20catalog/book/46
Therefore, error propagation can also be applied to the median filter, S3.2, when the smoothed value is obtained as a linear combination of existing values (with uncertainties) within the sliding window.

> We have added a sentence about how error propagation with the median filter would be possible when considered as a linear combination of the input quantities (Section 3.2):

> When applying error propagation on the median filter, the output value can be seen as a linear combination of the input values multiplied by weights $w \in \{0, 1\}$, where a single value has weight 1 and all other values have weight 0. Consequently, any uncertainty in the measurement of that single value directly propagates into the output of the median, and it is independent of the number of the points in the window and the window size.

The explanations on page 12 could be made more clear:
- R278 introduces a measurement noise matrix 'R', but this matrix is not used in the consecutive explanation. Would be good to see 'R at work' in some equations.

> We have added the Kalman prediction and update equations (Eqs. 8-13) and added a paragraph in the text. Consequently, we introduce the additional matrices $K, \bar{x},$ and $\bar{P}$ (each for a specific time step $t + \Delta t$):

> Running the Kalman filter then results in estimates of the state and its uncertainty for each point in time, based on all previous states and measurements. This is referred to as a "forward pass", as calculation on a

time series starts with the first measurement and then continues forward in time (Gelb et al., 1974, p. 156). The forward pass is given by the prediction (Eqs. 8-9) and update equations (Eqs. 10-13) (Labbe, 2014, Chapter 6):

First, we predict the future state $\bar{x}_{t+\Delta t}$ (at $t + \Delta t$) from the current state $(x_t)$ and the linear state transition $F$. The bar on $\bar{x}_{t+\Delta t}$ signifies a prediction without an update:

$$\bar{x}_{t+\Delta t} = F x_t \quad (8)$$

Next, we predict the future covariance by using the law of error propagation (cf. Eq. 2) and add the process noise $Q$. Here, the appropriate version of $Q$ is used, depending on the order of the model (either $Q_{xva}$, $Q_{xv}$ or $Q_x$).

$$\bar{P}_{t+\Delta t} = F P_t F^T + Q \quad (9)$$

In consequence, the state of the system becomes less certain over a longer time and can be made more certain by introducing a new observation with adequate uncertainty. For example, in the case of near-continuous TLS, change can be estimated one day into the future after having acquired one week of hourly observations. This allows estimating whether a larger interval between the observations still fully represents the expected changes.

If an observation at a given point in time $t + \Delta t$ is available, we subsequently update the state and state covariance. If not, Equations 8 and 9 are repeated for the next time step. Otherwise, for the update step, we first calculate the residuals $y_{t+\Delta t}$ by projecting the predicted state $\bar{x}_{t+\Delta t}$ into the observation space using the measurement function $H$ and subtracting this from the observations $z_{t+\Delta t}$:

$$y_{t+\Delta t} = z_{t+\Delta t} - H \, \bar{x}_{t+\Delta t} \quad (10)$$

Using the predicted covariance, the measurement function $H$, and the measurement noise $R_{t+\Delta t}$ (corresponding to the observation $z_{t+\Delta t}$), the so-called Kalman gain matrix $K_{t+\Delta t}$ is calculated:

$$K_{t+\Delta t} = \bar{P}_{t+\Delta t} H^T \, ( H \, \bar{P}_{t+\Delta t} \, H^T + R_{t+\Delta t})^{-1} \quad (11)$$

Finally, the Kalman gain $K_{t+\Delta t}$ is used to update the state vector by applying it on the residuals $y_{t+\Delta t}$ and adding to the predicted state $\bar{x}_{t+\Delta t}$:

$$x_{t+\Delta t} = \bar{x}_{t+\Delta t} + K_{t+\Delta t} y_{t+\Delta t} \quad (12)$$

Similarly, the state covariance is updated through the Kalman gain. Note that the term $KH$ is subtracted from the Identity matrix $I$, corresponding to decreasing values in the state covariance:

$$P_{t+\Delta t} = (I - KH) \, \bar{P}_{t+\Delta t} \quad (13)$$

Repeating Equations 8-13 for each time increment and measurement results in a set of state vectors $x_t$ and corresponding covariances $P_t$, which are a filtered and interpolated representation of the input time series. Each state is influenced by the previous states and measurements, as well as their corresponding uncertainties.

- R280 introduces Q, in the following, Q_{xva}, Q_{xv} and Q_x are used, but never properly introduced.
- Similarly, \sigma_a^2, and \sigma_v^2 are never introduced (in general: explain all symbols)

- R287: mentions \sigma^2 and refers to Eq. (6), but there is no \sigma^2 (without subscript) in Eq. (6).

We have fixed these points along with including the Kalman filter equations (see previous point).

What is the relation between 'discrete white noise" (R 286) and the following?

The process noise matrices given in Eqs. (5-7) represent discrete white noise models. We use this model to represent how the uncertainty of the state vector changes over time. We make this clearer now:

A common approach to model process noise is discrete white noise. Here, the variance of the highest-order element (e.g., the acceleration) is defined as $\sigma_a^2$ ($\sigma_v^2$ or $\sigma_x^2$ for lower-order models). The effect of this variance on the other elements of the system's state (i.e., velocity and position) is calculated following Equation 5 (Labbe, 2014, Chapter 7). We adopt this practice for our method.

What do you mean by 'highest order"? Order in the sense of polynomials?

Yes, a higher order model refers to terms of higher exponents in the polynomials. We have added this information to the text:

Note that we present a model of order 2 here (including position, velocity, and acceleration, i.e., a quadratic term as the highest-order term in the polynomial). If velocities are assumed constant, allowing for an acceleration term in the Kalman filter will lead to overfitting. Therefore, models of lower order may be considered (e.g., order 1, where velocity and position are modelled, or order 0, where solely position is modelled). In the case of $F$, the respective $n \times n$ submatrix in the top left corner can be extracted for these cases (where $n - 1$ is the order of the model).

In the presentation of the results, I would not present "linear interpolation" as (yet) another method, but as the raw time series

We have followed this suggestion and present it as the raw time series in the results, especially in Figs. 5 and 6, as well as throughout text.

F4: would be good to also include the original time series ("linear interpolation"). In addition, I cannot see any dotted lines in my A4 color print-out, the level of detection is not visible in the print out.

The raw time series has been added to Fig. 4, and we have increased the width of the dotted lines. A printout on an office printer confirmed that they are now visible:

**Comparison of Kalman filter models**

**a) Order 0 model (x)**

[Figure]

**b) Order 1 model (x,v)**

[Figure]

**c) Order 2 model (x,v,a)**

[Figure]

Extend the explanation of 'reduced level of detection' due to Kalman filter approach (seems to be an important result)

We have extended the explanation of the reduced Level of Detection and now present the significance for potential applications:

The exploitation of temporal autocorrelation, i.e., aggregating multiple measurements from multiple points in time yields much smaller Levels of Detection in the change detection. Consequently, smaller-magnitude changes - so long as they are permanent enough for the Kalman filter to pick them up - can be detected as significant. In the case of a slope movement, this is especially important, as small movements over long timespans add up to larger displacements. Through continuous observation, the detection of surface displacement as well as the quantification of the change rate can be achieved earlier, and with higher precision.

P368: intersection at "change of curvature" point: bug or feature? Is this good or bad to have?

We now elaborate on this:

This is a result of the minimization constraint: Assume that the trajectory before and after the change event is constant. As the RTS smoother minimizes the squared sum of residuals in observation space and constrains the change of the trajectory (whether it be displacement, velocity, or acceleration) at the same point, the resulting best estimated trajectory must be point-symmetric around the location of maximum residuals. As continuity is required by the physical model, the change of curvature must be at that exact location. For applications, the point of changing curvature in the smoothed time series can be leveraged to temporally locate sudden changes.

P376: please add a reference for the ringing artifacts.

We have added the following citation, which presents ringing artifacts at edges when using a space-invariant Kalman filter on image data. They propose a method to suppress this ringing based on applying a threshold and only using the Kalman filter before or after the edge, exclusively. This is similar to your suggestion raised in the Discussion re: "sudden changes".

A. M. Tekalp, H. Kaufman and J. W. Woods, "Edge-adaptive Kalman filtering for image restoration with ringing suppression," in IEEE Transactions on Acoustics, Speech, and Signal Processing, vol. 37, no. 6, pp. 892-899, June 1989, doi: 10.1109/ASSP.1989.28060.

I feel for F5 and F6, three different locations should be enough: a) is similar to e) and somehow to d); that would also allow to merge F5 and F6, by showing the results now in F5 and F6 for say, a) c) d) together (directly below each other).

We have merged Figures 5 and 6 into one figure, dropping the locations "b) Avalanche area" (where little to no movement is recorded during the investigation period) and "e) Boulder" (where only a single, sudden change occurs, which is similar to a)). The new locations have been mapped to a-c, and Fig. 1 has been updated accordingly.

[Figure]

398: "gold standard": for local validation the Total Station data could be used. Future work could also include the Kalman predicted displacement for next epoch with the measured displacement (as another way of validation)

We added the suggestion for future work to include point-based verification via total station measurements in the text:

For the real dataset, there is no validation data or other area-wide reference data with a much higher accuracy available, as TLS is considered to be the "gold standard'". Local, point-based validation could be achieved with total station measurements, if such measurements are available within the area of interest. In our case study in Vals, total station measurements were only available for reflectors installed in stable parts outside of the area of interest. This means that we cannot investigate whether the detected change is actual change for this dataset.

On the suggestion of comparing the Kalman predicted displacement to the next measurement: This is essentially being done within the Kalman filter, where residuals $y_{t+\Delta t}$ are calculated. The issue here is that these residuals contain both the measurement noise and the prediction error. For periods of little or no change (which is mostly the case for the Vals dataset), the prediction error is small compared to the measurement noise. Therefore, the quantified values would mainly relate to *a posteriori* variances of the bitemporal displacement values.

We now suggest to use the Kalman residuals as a way to identify parts of the dataset where the model is not a good approximation to the real change in the discussion (see comment regarding sudden changes below).

F9 and F10 are shown but hardly discussed, do they really contribute to the manuscript?

Figs. 9 and 10 show the performance of the Kalman-Filter on the virtual dataset, allowing for a validated interpretation of the results. We therefore believe they are important to be included in the manuscript, and have opted to present them in more detail:

In addition, we show the raw timeseries in comparison to temporal median smoothing for three locations (zero displacement and large positive/negative displacement) in Figure 9. At the locations in Figures 9a) and 9c), the true displacement amounts to -4 and +4.5 cm, respectively. Here, the Kalman filter estimates provide a much smoother trajectory than the temporal median, but tend to cut short at the curvature extremes around epochs 5 and 33, where the true change is over- or underestimated. In the case of Figure 9c), the overestimation of change is continued into epoch 40 by all methods, as measurement noise is too large. At the location shown in Figure 9b), the true change value is zero, and all derived displacement values (shown as the raw timeseries in black), are solely due to noise. At the end of the time series, there are a few epochs that indicate a positive change of around 3 mm, a trend which the two temporal median filters follow. The Kalman estimates, however, stay within 1 mm of the true value and are correctly below the Level of Detection.

The detected change at the end of the simulated 40-day change process is shown in Figure 10, where the different levels of detection result in a large difference in terms of detectable change. The bitemporal change detection using M3C2-EP only detects changes at the very border of the planar surface as significant, where a change magnitude > 4 cm is observed (note that the point density and change magnitude were designed to demonstrate this). By using the Kalman filter on the full time series, the trajectory is smoothed and denoised, leading to a lower Level of Detection. As a result, changes above approx. 8 mm can be confidently and robustly detected. Given only the data shown in Figure 10a, it is difficult to model the true change of the underlying surface. With the data presented in Figure 10b, a spatial deformation model can be fitted, allowing to extract the real change we applied to the original mesh model with much higher precision.

Looks like there is no link from the text to Fig. 8. Either remove the figure or discuss it (I would suggest to discuss it)

We added the missing link to Fig. 8a, now both subfigures are referenced. We have extended the discussion to these results:

In Figure 8a, we show the actual detected change and its magnitude for each core point at the end of the time series, i.e., after the avalanche event. The deposition of snow in both the avalanche area (with magnitudes > 10 cm) as well as generally in the lower slope area (with magnitudes from 2 to 6 cm) are clearly visible. In addition, significant negative change is observed on the right-hand side in blue, where vegetation is present in the dataset. Comparing Figure 8a with Figure 1b visualizes the benefit of the multitemporal approach, as it is able to correctly detect much more change as significant than the bitemporal one.

We show the locations of points where change is detected with only the bitemporal approach, where it is detected with multitemporal Kalman filtering, and where it is detected with both in Figure 8b. 26.92% of the core points in the study area were attributed with significant change when using the multitemporal approach but not with the bitemporal approach. This mostly concerns areas on the lower slope (colored in blue), where the magnitudes are between 0.02 and 0.06 m from deposited snow. In contrast, about 4.26% of the core points were attributed with significant change in the bitemporal analysis, but not for the multitemporal case.

In general, further focus your results, e.g. by elimination: First, choose your best settings for the time series processing; Second, show the improvement in change detection between best settings and traditional pair-wise M3C3, Finally, show some different results for k, but only using the best settings; The results of the simulated data set could also be moved to the discussion -> are both Figure 11 and 12 needed? Could be combined into one, showing the most interesting results

The results section follows the path:
- Finding the optimal variance values for the Order 0, 1, and 2 model
- Comparing these models to find the best model, comparing it with the state-of-the-art (raw timeseries and temporal median) on real data
- Showing these results on virtual data, which allows us to quantify the improvement over the state-of-the-art (rather than giving visual impressions)
- Presenting how subsequent processing (i.e., clustering) is affected by the choice of time series filter.

We have outlined this path clearer in the first paragraph of the results section:

We first present the impact of different model and parameter choices on the clustering and change detection results (Sect. 4.1), and select an optimum variance value ($\sigma^2_{x/v/a}$) for each of the Kalman models. Subsequently, we compare our results with the raw time series, temporal median smoothing, and bitemporal M3C2-EP (Sect. 4.2) for both selected locations and for the whole area of interest. To quantify the actual improvement in terms of residuals, we present the results of the synthetic experiment in Section 4.3. Finally, clustering is carried out on the filtered time series with the best parameter settings on the real data, the results of which we show in Section 4.4.

We believe that both Figs. 11 and 12 are important, respectively: In Fig. 11, we present the result for different choices of k (a specific request from the first review), which is critical when applying K-Means clustering (also following Kuschnerus et al., 2021).
Figure 12 showcases results based on our method when compared to the state-of-the-art. There is, however, no need to show the results of the Order 0 and the Order 1 model, as we have eliminated them earlier on. Therefore, we remove Figs. 12 a and b and just present the result of the temporal median model and the raw timeseries, with 10 clusters each. The manuscript text has been adapted accordingly.

Would be interesting to see the average Kalman velocity (acceleration) per core point.
We plotted the average filtered velocity and acceleration (both taken from the "best" Order 2 model) below:

[Figure]

a) Mean velocity (Order 2 model)   b) Mean acceleration (Order 2 model)

The area of the avalanche and erosion at the edge of the area of interest (cf. Fig 1 b) are well visible. The velocity model does not give much more insights than the detected change as shown in Fig. 8 b), but the acceleration shows an interesting pattern where the lower slope of the area of interest is subject to mean negative accelerations in the range of >1/2 mm per day squared. Looking at the acceleration time series at one of these locations (marked with an X in Subfigure b) shows that this corresponds mainly to snow accumulation and melt at the end of the period:

[Figure]

Point ID 368836

Below, we show the mean velocity and acceleration per core point on the time series cut off before the avalanche event (at t=1850 hours):

[Figure]

Finally, we show the mean velocity and acceleration per core point on the time series cut off before any snowfall (at t=1750 hours):

[Figure]

For reference, we show the time series for the same core point as before (note the change in scale):

[Figure]

We can see how the mean values for velocity and acceleration are largely influenced by the sudden events towards the end of the time series. Furthermore, at this point in time, the Order 1 and especially the Order 2 models overfit and exhibit ringing artefacts, and the Order 0 model provides a much better trajectory.

We opt not to include these plots in the manuscript for the following reasons:
- The Order 0 model, (providing the best trajectory for this core point) does not include an estimate of velocity or acceleration (other than through numerical integration). While we chose a model of order 1 for our dataset globally, this still does not contain acceleration values.
- The average values for each core point seem to be very dependent on the temporal window that is selected.
- We originally included velocity and acceleration plots for selected core points, but removed them following reviewer comments in the initial iteration.

I would move the percentages of change locations (4.26 % in r 441) to the results

We have removed the exact number from the discussion but have added a link to the relevant Section where it is already presented (end of Section 4.2):

The number of locations that were detected using the bitemporal M3C2-EP method, but not when using the multitemporal approach (cf. Section 4.2), is close to the theoretical number of false positives (5% when using a level of significance of 95%), when considering that some of these 5% of false positives will again be false positives (i.e., incorrectly identified) in the multitemporal method using the same level of significance.

"The Kalman filter is ill-suited to represent sudden changes": possible future work could be to test if a new measured value fits the Kalman trend given the confidence in both the trend and the measured value, this could result in a kind of trend change detection,

This suggestion is now included in the discussion:

Furthermore, future research could attempt to detect sudden changes in the time series, e.g., by analysing the Kalman residuals (cf. Eq. 10) in relation to the measurement and prediction confidences.

Kalman filtering has been previously used to smooth bathymetric MBES data, e.g. Bourgeois, B. S., Elmore, P. A., Avera, W. E., & Zambo, S. J. (2016). Achieving comparable uncertainty estimates with Kalman filters or linear smoothers for bathymetry data. Geochemistry, Geophysics, Geosystems, 17(7), 2576-2590.

We have added this relevant citation in the Introduction:

Kalman filters are commonly used today in trajectory estimation, e.g., for direct georeferencing of airborne laser scanning data (El-Sheimy, 2017). They have also been used for bathymetric uncertainty estimation in hydrographic applications using multi-beam echo sounding data (Bourgeois et al., 2016).

MINOR POINTS:

Abstract: "almost double", relative to what baseline method?

The values refer to the bitemporal M3C2-EP analyses, as shown in Fig. 8b . We add this information to the abstract:

The method enables to almost double the number of points where change is detected as significant (from 24% to 47% of the area of interest), compared to bitemporal M3C2 with error propagation.

Abstract: "this can be a critical': can be positive or negative, maybe reformulate as 'This is a solution for subsequent analysis methods that…"

Adapted as suggested.

Intro, r28: "needs to" -> "is" (needs to would require a mathematical proof that there is no other way)

Adapted as suggested.

Intro, r48: "detectable" -> "detecting"

Adapted as suggested.

Caption, F2a): -> "Histogram of the number of points found"

Fig. 2a was removed (see major points), therefore no longer applicable.

S3.3., r250: I would call it an "assimilation" method, rather than an "online" method.

Adapted as suggested.

S3.3, r250: "post hoc" -> "a posteriori"

Adapted as suggested.

R219: 'right' -> left

Error has been resolved.

R247: 'empliyed'

Changed to 'employed'

R259: -> "For a state vector $x\_t$ containing…"

We have added the symbol $x\_t$.

R263: -> The diagonal entries of Eq. (3) ensure"

Adapted to:

The diagonal entries of Eq. 3 with value 1 ensure…

R280: skip "Finally"

    Adapted as suggested.

R305: -> The impact of the exact choice"

    Adapted as suggested.

R328: -> "time series similarity"

    Adapted as suggested.

R328: -> "How the differently smoothed time series" (right?)

    Correct - Adapted as suggested.

R331: -> "Euclidean"

    Error has been resolved.

Ch 4: in the first lines, I would explicitly introduce S4.1 to s 4.4

    We have introduced the subsections explicitly in the first paragraph of Section 4 (see response to previous comment)

caption , F10: "larger" -> "starting from"

    Adapted as suggested.

---

## Author Response (AR3)

**AUTHOR'S COMMMENTS**

Dear Prof. Hovius,

thank you for the positive response on the revisions we have carried out on our manuscript. We believe that the reviewer's comments have substantially improved the quality of our work, for which we are very grateful.

Please find the final changes made to the 2$^{nd}$ revision in response to the comments below.

We have further replaced the reference to Hartl et al. (2019) by a more recent reference to Berger et al. (2021), who give a slightly different estimate of the rockfall volume (117,000 m³ rather than 116,000 m³). Apart from that, we fixed minor typographical and grammatical errors, which you can find in the difference document.

On behalf of all authors,

Lukas Winiwarter

**DETAILED RESPONSE TO REVIEWER COMMENTS – REVIEWER #1**

Minor remarks:
1. r42: singular → single (singular values belong to Linear Algebra)
2. r242: → 'number of points' (remove 'the')
3. r266: → 'implementations of the methods'
4. r324: noide →noise
5. r452: 'so long' → 'as long'
6. Caption, Figure 6: 'temproral' → 'temporal'
7. Caption, Fig. 6: → 'window size' (remove 's')
8. r462: 'either' → 'both'
9. Caption, Figure 9: 'negative maxima': is that not the minimum?
    Changed to 'negative and positive extrema'

10. Caption, Fig. 10: 'still reliable' → 'consistent'
    We carried out all changes as suggested. Thank you for taking the time to improve our manuscript.

**DETAILED RESPONSE TO REVIEWER COMMENTS – REVIEWER #3**

I think this paper offers a very interesting contribution to the journal, and i see the authors provided a thorough review of the paper addressing all the raised points. i only have few technical comments that could be addressed

L475
The exploitation of oral autocorrelation … AS LONG AS… rather than so long as*

    Changed as suggested.

Fig 4, it would be best to reference it before the figure itself

    In the final version (i.e., not the difference document), the reference for Fig. 4 appears before the figure itself.

Median filter: what's the chosen window? I see you refer to it in the results and discussion ("using a window size of 96 hours") but I would suggest to report this in the method chapter directly.

The window size of the median filter was added to the methods section:
In our application, we chose 96 hours as the window size in order to remove any daily and sub-daily effects.

Fig 7 could you please highlight in the figure the area where the erosion rills are, for clarity? The figure shows lots of area with >50%, it would be easier to catch the reader attention to the one you point out in the caption

We marked the location of the erosion rills in Figure 7:

[Figure]

Figure 9: maybe I missed it in the text, if so i apologies, but what's the cause of high measurement noise after epoch 40?

Figure 9 ends with Epoch 40, and there is no obvious increase in measurement noise in Figure 9. Significant increase in measurement noise is observed in Figs. 5 and 6a, where alignment errors lead to increased uncertainty. We have added this explanation in the text:

Note how especially in the last third of the time series, with a larger variance in the raw time series, models with larger values for $\sigma_x$ tend to fit the oscillations rather than smoothing over them. The cause for the increased uncertainty is the location of the investigated point, which is on the edge of an erosion rill. Dependent on the alignment of individual epochs, the proportion of points within the M3C2 cylinder that have changed varies, resulting in the visible oscillating pattern.

Figure 10, it seems a bit confusing that this figure is in the chapter related to the clustering analysis, while it is discussed and referenced in the previous subchapter.
Similarly, 11 and 12 are in the discussion, but they are addressed previously.

We have optimized the positioning of Figures by introducing additional formatting constraints in the manuscript. Figure 10 is now located within Section 4.3, and Figs. 11 and 12 are located in Section 4.4.

Line 565 pag 27, why the question mark?

This was an incorrectly formatted reference in the difference document to Kromer et al. (2017), which has been resolved to:

[revised manuscript text omitted]